# User-Aware Active Knowledge Acquisition for Emotional Support Dialogue

**Mufan Xu** [1]   **Kehai Chen** [1]   **Jiahao Hu** [2]   **Xinchao Xu** [2]   **Muyun Yang** [1]   **Tiejun Zhao** [1]   **Min Zhang** [1]

## Abstract

Emotional support plays an important role in dialogue systems, and its success depends on adapting to a user's evolving and implicit needs across multi-turn interactions while leveraging the strong reasoning capacity of large language models. However, since signals about user needs are often weak, indirect, and can only be disambiguated through multi-turn interaction, existing emotional support methods often struggle to acquire and generalize relevant conversational knowledge efficiently. To bridge this gap, we introduce **User-Aware Active Knowledge Acquisition (UKA)**, a gradient-free active dialogue learning framework that explicitly represents uncertainty about user needs and incorporates active learning into both knowledge acquisition and response selection. We propose a Theory-of-Mind uncertainty estimation mechanism that allows the model to prioritize responses, thereby eliciting more informative user feedback. UKA is capable of efficiently exploring user-aligned conversational knowledge during training while maintaining robustness at test time. Experiments across multiple dialogue benchmarks and model architectures demonstrate that our approach consistently outperforms strong baselines in dialogue quality and user alignment.

## 1. Introduction

Large language models (LLMs) have rapidly advanced the quality of dialogue systems, enabling strong instruction following, multi-turn coherence, and broad task coverage (Meng et al., 2024; Zhang et al., 2025b;d). In emotionally sensitive settings, beyond fluent generation, success depends on emotional intelligence (EQ): recognizing affect, validating feelings, and adapting support

strategies to a user's implicit and evolving needs over multi-turn interaction (Wang et al., 2024; Tang et al., 2025). These emotional-support needs are incrementally clarified through interaction, where the assistant's responses shape what evidence the user reveals next (Dongre et al., 2025). Recent benchmarks and agent frameworks further stress user alignment and social-cognitive competence, suggesting that effective emotional support requires dynamic, user-specific adaptation (Kim et al., 2025; Zhang et al., 2026).

However, user needs in emotional support are often weakly signaled and only clarified through multi-turn interaction, making it difficult for existing methods to efficiently acquire and generalize emotionally relevant knowledge. Generic supportive responses or standalone intent inference can therefore appear helpful while mismatching the user's actual need, leading to pushback when a different form of support is expected (Figure 1 (a)). Meanwhile, strategy-memory and agent-style planning methods (Zhang et al., 2026; Kim et al., 2025; Zhang et al., 2025c) often follow passively acquired LLM behaviors, yielding redundant feedback and limited coverage of the EQ knowledge space (Figure 1 (b)). Furthermore, storing EQ knowledge in an action-agnostic or user-action-only form ignores that user feedback depends on the assistant's chosen response, so similar surface emotions may require different user-aware strategies (Figure 1 (c)). These limitations motivate an interaction-centric framework that actively selects responses to both support the user and elicit diagnostic feedback for EQ knowledge acquisition.

To address these issues, we propose **User-Aware Active Knowledge Acquisition (UKA)**, a knowledge-driven and *gradient-free* framework that treats emotional support dialogue as both a support process and an active knowledge-acquisition process under uncertain user needs. In particular, UKA (i) performs *user-aware* EQ knowledge retrieval by constructing a summary anchor that reflects both dialogue context and user-need uncertainty, and (ii) casts response selection as an *uncertainty-driven active learning* problem: it favors candidates with high ToM-based diagnostic uncertainty and low knowledge support to efficiently expand the EQ memory during training, while at test time it prioritizes knowledge-grounded responses that reduce expected user-model uncertainty for stable support. This separation enables efficient knowledge acquisition without policy updates or gradient-based adaptation, and yields

[1] Harbin Institute of Technology, China [2] Baidu Inc., Beijing, China. Correspondence to: Kehai Chen <chenkehai@hit.edu.cn>.

*Proceedings of the 43 $^{rd}$ International Conference on Machine Learning*, Seoul, South Korea. PMLR 306, 2026. Copyright 2026 by the author(s).

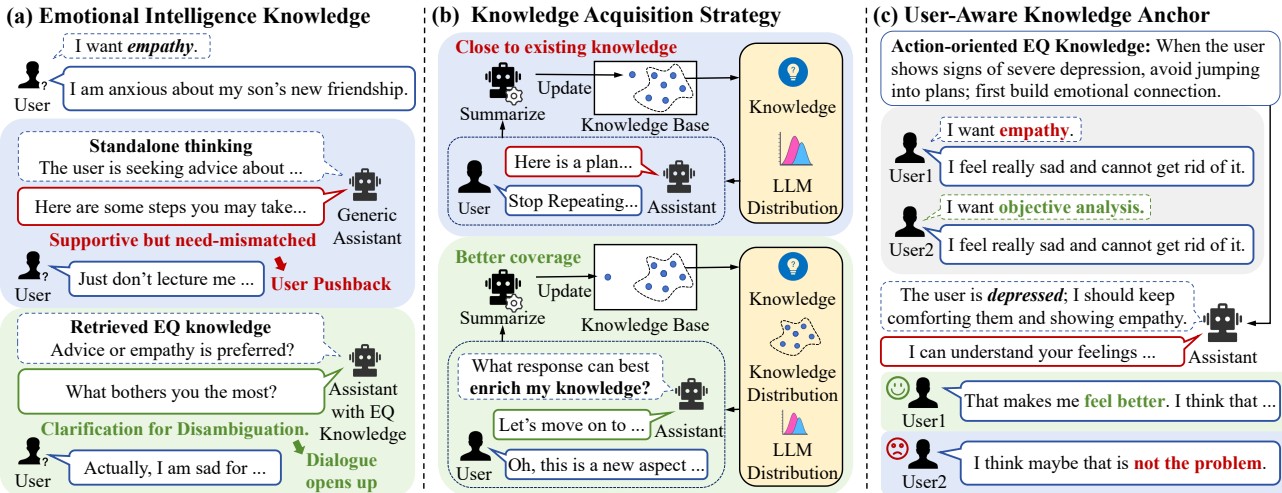

*Figure 1.* Motivation for User-Aware Active Knowledge Acquisition in emotional support dialogue. (a) EQ knowledge may help clarify a user's needs, which are often implicit; generic supportive replies can mismatch the need and trigger pushback. (b) If interaction stays close to existing knowledge, the system collects redundant signals; actively selecting responses yields better coverage. (c) The same surface emotion may reflect different user preferences, requiring user-aware knowledge and response selection to achieve alignment.

robust, emotionally appropriate behavior under distribution shift and evolving users.[1] Our main contributions are:

- We introduce **UKA**, a gradient-free active dialogue framework that separates exploratory EQ knowledge acquisition from test-time knowledge use.
- We propose a **ToM-based diagnostic uncertainty** mechanism that tracks user-need hypotheses and guides response selection via hypothesis-conditioned reaction disagreement.
- We show that UKA improves over prompting, memory-based, and ToM-style baselines on three emotional support benchmarks and multiple LLM backbones.

**Conflict of interest disclosure.** The experiments in this paper do not evaluate models or products developed by the authors and their affiliations. To the best of our knowledge, there are no financial conflicts of interest that could reasonably be perceived to influence this work.

## 2. Related Works

### 2.1. Knowledge-Augmented LLM Systems

LLM-based systems often require external knowledge to improve freshness and faithfulness, making retrieval-augmented generation a standard grounding recipe (Gao et al., 2023; Cheng et al., 2025a; Xu et al., 2025). Recent works emphasize *multi-turn* conversational RAG, including datasets/benchmarks and pipelines that study query rewriting, when-to-retrieve decisions, and grounded response generation (Choi et al., 2024; Cheng et al., 2025b; Roy et al., 2024), as well as training under limited supervision via synthetic grounded dialogues, latent-

variable grounding, and LLM-based augmentation for conversational dense retrieval (Chen et al., 2024a; Ye et al., 2024). To quantify and mitigate hallucination in grounded dialogue, recent evaluation resources provide RAG hallucination corpora and dialogue-level hallucination benchmarks, complemented by claim-level factuality scoring and reference-free RAG evaluation toolkits (Niu et al., 2024; Chen et al., 2024b); meanwhile, memory augmentation is increasingly treated as another grounding channel, with dedicated benchmarks and multilingual meta-evaluation suites (He et al., 2025; Cruz Blandon et al., 2025).

### 2.2. Active Learning in NLP

Active learning (AL) in NLP typically aims to reduce annotation cost by selecting informative instances (Settles, 2009; Lewis & Gale, 1994). For interactive systems, supervision often comes as implicit user feedback, shifting AL toward online decision making under partial feedback; this is commonly formalized with contextual bandits or dueling feedback to balance exploration and exploitation (Yue & Joachims, 2009; Liu et al., 2018). In the LLM era, such interaction-driven learning is closely related to preference optimization RLHF-style post-training (Rafailov et al., 2023; Yan et al., 2026), including group preference objectives such as GRPO (Shao et al., 2024), while iterative RLHF is used to cope with drifting user distributions (Dong et al., 2024) despite challenges like feedback noise and distribution shift (Casper et al., 2023). In our setting, active learning is not used to select instances for annotation; instead, it selects the *assistant reply* that makes the *user feedback* most informative for EQ knowledge acquisition, which is closely related to active prompting for maximizing learning signal from limited interactions (Diao et al., 2024).

---

[1]Code available at github.com/Xmuffins/UKA.

## 2.3. User Modeling and Theory of Mind

User modeling in dialogue maintains a latent belief over what a user prefers and will do next, so that the agent can plan responses that are not only helpful but also informative for future interactions. Recent personalization works model this belief via profile/persona conditioning and memory retrieval, including retrieval-augmented personalization to compensate for sparse user profiles (Huang et al., 2023) and benchmarks that stress-test long-horizon interactive memory behaviors (Wu et al., 2025; He et al., 2025), while other lines update explicit user representations directly from feedback (Sun et al., 2024; Zhang et al., 2024). Complementary to this, Theory of Mind (ToM) views conversation as reasoning about others' beliefs and intentions, which is increasingly evaluated and modeled in LLM-based agents (Strachan et al., 2024; Zhu et al., 2024); beyond static tests, recent methods treat ToM as a controllable inference-time capability and as an agent property measurable in agent–agent interaction (Cross et al., 2025; Choi et al., 2025; Yang et al., 2025). In our setting, the user belief state guides reply selection to elicit informative feedback, with LLM-based user simulation enabling scalable and controllable feedback-driven updates (Davidson et al., 2023).

## 3. Problem Formulation

We study knowledge-driven multi-turn emotional support dialogue under *uncertain user needs*. At each turn, the assistant must (i) provide a helpful response and (ii) interact in a way that improves its ability to infer the user's latent need and reuse emotionally effective knowledge.

**Dialogue setting.** A dialogue is a sequence of turns $\mathcal{H}_t = \{(r_1, x_1), \ldots, (r_t, x_t)\}$, where $r_i$ and $x_i$ denote the assistant response and user reply at turn $i$. At turn $t+1$, the assistant generates $r_{t+1}$ conditioned on $\mathcal{H}_t$. Each dialogue is conducted between two models: an assistant model and a user simulator that simulates user responses. In this work, we optimize only the assistant model; the user simulator is fixed and serves as part of the interaction environment.

**EQ knowledge memory.** The system maintains an external memory $\mathcal{K}$ of EQ knowledge entries (e.g., reusable emotional strategies). At turn $t$, the system retrieves a small subset $\mathcal{K}_t \subset \mathcal{K}$ to ground response generation. During training, $\mathcal{K}$ can be expanded by writing new entries from interaction feedback; at test time, $\mathcal{K}$ is fixed. For all embedding-related processes, we employ the EmbeddingGemma-300M model (Vera et al., 2025).

**User belief and hypothesis set.** The user simulator generates replies according to an underlying need $u^*$. The assistant neither observes $u^*$ nor updates the user simulator. Instead, it maintains an assistant-side user belief, consisting of a hypothesis set $\mathcal{U}_t = \{u_t^{(1)}, \ldots, u_t^{(M)}\}$ and a belief

distribution $p_t(u|\mathcal{H}_t)$. Each $u_t^{(i)}$ is a natural-language description of a plausible user need or interaction goal. After each user reply, only this belief state is updated.

**Candidate responses and selection.** At each turn, the model generates a candidate set $\mathcal{C}_t = \{c_t^{(1)}, \ldots, c_t^{(N)}\}$ conditioned on $\mathcal{H}_t$ and retrieved knowledge $\mathcal{K}_t$. Response selection is a phase-dependent decision rule

$$r_{t+1} = \pi_{\text{phase}}(\mathcal{C}_t; \mathcal{H}_t, \mathcal{K}_t, p_t), \quad (1)$$

where $\pi_{\text{phase}}$ uses the current belief $p_t$ and knowledge grounding to rank candidates.

**Training vs. test time.** In Eq. (1), the phase determines whether selection is used for knowledge acquisition or knowledge usage. Training favors exploratory responses that elicit informative feedback to expand $\mathcal{K}$, whereas test-time selection uses the fixed memory for stable support, with only belief updates and no parameter or memory updates.

## 4. Method

Based on the formulation in Section 3, we propose a gradient-free framework for acquiring and utilizing EQ knowledge under uncertain user needs.

### 4.1. EQ Knowledge Base: Representation and Update

We define EQ knowledge as actionable, reusable conversational guidance that helps an assistant respond appropriately to a user's affect, implicit needs, and interaction intent. Unlike factual knowledge, EQ knowledge is evaluated by its *interaction effectiveness* rather than objective truth. We maintain an external memory $\mathcal{K} = \{k_1, \ldots, k_L\}$ of EQ knowledge entries. Each entry $k$ is stored in a *state–action* form $k = (a, v)$, where $a$ is a retrieval anchor describing an emotionally salient dialogue state along with the inferred user needs, and $v$ is the recommended response behavior or strategy.

**User-dependent relevance.** Given dialogue history $\mathcal{H}_t$, a candidate response $c$, and a user-need hypothesis $u$, we assume that only a subset of EQ knowledge is relevant:

$$\phi(\mathcal{H}_t, c, u) \subseteq \mathcal{K}. \quad (2)$$

Following the motivation in Section 1, EQ knowledge is interaction-dependent: its usefulness is determined by how well the supported response matches the user's latent needs and subsequent feedback, rather than by an isolated factual criterion. Instead of predicting emotion labels, we assess whether a response supported by retrieved EQ knowledge aligns with the latent user needs $\mathcal{U}_t$.

**Knowledge Acquisition.** New EQ knowledge is written after the system takes an action and observes user feedback.

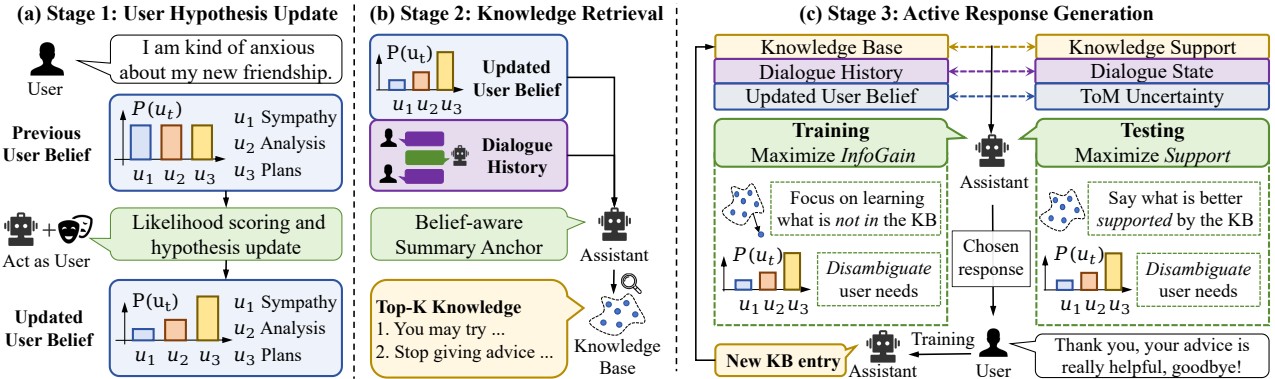

**Figure 2.** Method overview of UKA as a three-stage pipeline. (a) **User hypothesis update**: maintain a belief distribution over user-need hypotheses and update it by scoring the likelihood of the observed user reply under each hypothesis. (b) **Knowledge retrieval**: construct a belief-aware summary anchor from the dialogue history and current belief, and retrieve top-$K$ EQ knowledge entries from the external knowledge base. (c) **Response generation and selection**: generate candidate replies conditioned on the history and retrieved knowledge, estimate training-time exploration or test-time exploitation, select the best response, and summarize new knowledge for future retrieval.

We summarize the interaction into a new state–action entry $k = (a, v)$ and append it to $\mathcal{K}$, where $a$ is a belief-aware anchor describing the dialogue state and inferred needs, and $v$ is an actionable strategy distilled from the feedback.

## 4.2. Stage 1: User Belief Update

As shown in Figure 2(a), we treat the user's underlying need as latent and maintain an explicit hypothesis set. Let $u^*$ denote the true but unobserved user need. At turn $t$, we maintain a hypothesis set $\mathcal{U}_t = \{u_t^{(1)}, \ldots, u_t^{(M)}\}$, where each $u_t^{(i)}$ is a natural-language description of a plausible user need. We denote the dialogue history as $\mathcal{H}_t = \{(r_1, x_1), \ldots, (r_t, x_t)\}$, where $r_t$ is the assistant response and $x_t$ is the user reply at turn $t$. To measure how well each hypothesis matches the newest user reply, we compute a teacher-forced compatibility score using the LLM. We prepend the hypothesis description as a system instruction ("act as a user with need $u_t^{(i)}$"), concatenate it with the dialogue context including $x_t$, and obtain scores of the hypothesis using the same assistant model:

$$\ell_i^{(t)} = \frac{1}{|x_t|} \sum_{k=1}^{|x_t|} \log p(x_{t,k} \mid H_{t-1}, r_t, u_i), \quad (3)$$

where $|\cdot|$ is the token length. Intuitively, a higher $\ell_i^{(t)}$ indicates that the observed interaction is more self-consistent under hypothesis $u_t^{(i)}$. We then normalize these scores to obtain a belief distribution over hypotheses:

$$p_i^{(t)} = \text{softmax}_i\left(\ell_i^{(t)}\right), \text{ where } p_i^{(t)} \triangleq p\left(u_t^{(i)} \mid \mathcal{H}_t\right). \quad (4)$$

**Hypothesis expansion and refresh.** We refresh the hypothesis set when the current set is likely to be incomplete or ambiguous: (i) $|U_t| < M$; (ii) a newly proposed hypothesis obtains a higher compatibility score than the

best existing hypothesis, suggesting that the current set misses an important interpretation of the user need; or (iii) the belief entropy remains above a threshold, indicating persistent uncertainty. After generating a new hypothesis, we rescore all hypotheses using Eq. (4). If the number of hypotheses exceeds M, we remove the hypothesis with the lowest posterior probability.

## 4.3. Stage 2: Belief-Aware EQ Knowledge Retrieval

As shown in Figure 2(b), UKA retrieves EQ knowledge by conditioning the retrieval query on both the dialogue context and the current belief over user-need hypotheses.

**User-needs-aware summary anchor.** Given the dialogue history $\mathcal{H}_t$, the hypothesis set $\mathcal{U}_t = \{u_t^{(i)}\}_{i=1}^M$, and the belief vector $p^{(t)} = \{p_i^{(t)}\}_{i=1}^M$, we construct a retrieval query (summary anchor) via an LLM summarizer:

$$s_t = f_{\text{sum}}(\mathcal{H}_t, \mathcal{U}_t, p^{(t)}), \quad (5)$$

where $f_{\text{sum}}(\cdot)$ extracts emotionally salient signals and incorporates the most likely user needs, avoiding commitment to a single hypothesis under uncertainty.

**Top-$K$ retrieval from memory.** Recall that each EQ knowledge entry is stored as $k = (a, v) \in \mathcal{K}$ (Section 4.1), where $a$ is the retrieval anchor. We retrieve the top-$K$ entries whose anchors best match $s_t$:

$$\mathcal{K}_t = \text{Top-}K_{k=(a,v)\in\mathcal{K}} \text{ sim}(s_t, a), \quad (6)$$

where $\text{sim}(\cdot)$ is implemented by a cosine similarity. The retrieved set $\mathcal{K}_t$ is then used to ground candidate response generation and selection in Stage 3.

## 4.4. Stage 3: Uncertainty-Driven Active Response

The training and testing objectives of UKA differ in how they trade off diagnostic uncertainty and knowledge

support: training favors responses that expose missing knowledge, whereas testing favors responses supported by the acquired memory while retaining mild diagnostic value. As shown in Figure 2(c), given the retrieved EQ knowledge $\mathcal{K}_t$, the system generates candidate responses $\mathcal{C}_t = \{c_1, c_2, \ldots, c_N\}$ conditioned on the dialogue history and retrieved knowledge. Since the assistant's action shapes what evidence the user reveals next, response selection is central to both learning and support quality. We therefore formulate it as *uncertainty-driven active learning* with an explicit train–test separation: training probes and disambiguates the user's latent need to acquire diverse EQ knowledge beyond near-known behaviors, while testing favors *knowledge-grounded* replies that provide robust support under distribution shift and retain diagnostic value. This requires a unified scoring function that jointly measures (i) *knowledge support* and (ii) *ToM uncertainty*.

**Knowledge support.** To avoid writing and reusing EQ knowledge beyond what the current memory can reliably justify, we score each candidate by how strongly it is supported by the retrieved knowledge, providing an explicit estimate of the system's *knowledge boundary* at turn $t$. For each candidate response $c_j$, we estimate its knowledge support score with respect to the retrieved knowledge set:

$$S_k(c_j) = g(c_j, \mathcal{K}_t), \tag{7}$$

where $g(\cdot)$ is computed as the average cosine similarity between the candidate response and its corresponding top-k closest knowledge entries. Using the knowledge support score, UKA can effectively perceive the boundaries of acquired knowledge, thereby selecting responses that are better aligned with what it already knows.

**ToM uncertainty.** In emotional support dialogue, the user's next reply is *response-contingent*: different assistant actions can elicit different evidence even under the same latent need, making passive learning from observed user replies inefficient. We therefore treat each candidate response as an *active probe* and quantify how much it would induce distinguishable user reactions across competing user-need hypotheses—so that the system avoids repeatedly eliciting redundant signals near existing behaviors. To achieve this, we propose using a *Theory-of-Mind Uncertainty* to measure the distance between responses under competing user-need hypotheses. Concretely, for each $u_i \in \mathcal{U}_t$ we use a ToM simulator to generate a group of next-turn replies and obtain an average embedding $\mathbf{e}_i = \mathbf{e}(u_i, c_j)$. We then define ToM uncertainty as the average cross-hypothesis disagreement:

$$S_u(c_j) = 1 - \frac{1}{M}\sum_{i=1}^{M} \cos(\mathbf{e}_i, \boldsymbol{\mu}), \quad \boldsymbol{\mu} = \frac{1}{M}\sum_{i=1}^{M}\mathbf{e}_i, \tag{8}$$

where we use $M = |\mathcal{U}_t|$ to denote the size of the user-need hypothesis set. A larger $S_u(c_j)$ indicates that $c_j$

is more discriminative for disambiguating user needs, while a smaller value suggests the candidate mainly elicits redundant feedback.

**Training-time exploration.** During training, we aim to acquire diverse EQ knowledge with minimal interaction cost. The ToM uncertainty score $S_u(c_j)$ favors responses whose *action-conditioned* user reactions differ across competing need hypotheses, yielding more informative feedback for belief updating and knowledge writing. Meanwhile, subtracting the knowledge support $S_k(c_j)$ encourages stepping beyond the current knowledge boundary and avoids repeatedly collecting redundant signals:

$$c_t^{\text{train}} = \arg\max_{c_j \in \mathcal{C}_t} \left( S_u(c_j) - S_k(c_j) \right). \tag{9}$$

**Test-time exploitation.** At test time, the system prioritizes reliable support by choosing responses that are well grounded in the acquired EQ knowledge. When the user's need remains uncertain, it additionally prefers responses that are more *diagnostic* to quickly disambiguate the user state through the next-turn feedback:

$$c_t^{\text{test}} = \arg\max_{c_j \in \mathcal{C}_t} \left( S_k(c_j) + \gamma\, S_u(c_j) \right), \tag{10}$$

where $\gamma$ controls the trade-off between knowledge-groundedness and need disambiguation (default set to 1).

## 5. Experimental Results

### 5.1. Experimental Settings

**Models and Benchmarks.** We evaluate our framework on four large-scale open-source LLM backbones. We use two dense models, Qwen3-32B (Team, 2025) and Seed-1.6-36B-Instruct (Seed, 2025), and two MoE models, Qwen3-235B-A22B-Instruct (Team, 2025) and GPT-OSS-120B (OpenAI, 2025). We conduct experiments on three emotional support dialogue benchmarks ESConv, ExTES (Liu et al., 2021) and Sentient Eval (Zhang et al., 2025a) that reflect complementary aspects of emotionally intelligent dialogue. Sentient Eval is a Chinese-language benchmark; we conduct all experiments on it in Chinese and translate the examples into English for readability, with data statistics and benchmark details provided in Appendix A.1.

**Baselines.** We select three representative categories of methods as baselines: (i) **Prompting Baseline** serves as a minimal yet strong baseline that encourages the model to act as a psychotherapist, reflecting the performance of standard LLM interaction. (ii) **PRINCIPLES**-style baseline follows the synthetic strategy-memory framework of Kim et al. (2025); in this paper, **PRINCIPLES** refers to our controlled implementation of this baseline. We view it as a basic form of memory-level self-evolution, where reusable dialogue

*Table 1.* Results of UKA and baseline paradigms across multiple open-source models. Bold numbers indicate the best performance under the same model and dataset, while underlined numbers denote the second-best results.

| Models | Methods | ESConv | | ExTES | | Sentient Eval | | | |
|---|---|---|---|---|---|---|---|---|---|
| | | SR↑ | AT↓ | SR↑ | AT↓ | Avo.↑ | Pas.↑ | Neg.↑ | *Avg.↑* |
| Qwen3-32B | Prompting Baseline | 0.306 | 7.93 | 0.510 | 7.32 | 44.1 | 31.9 | 16.1 | 28.5 |
| | *w.* MetaMind | 0.338 | 7.97 | 0.533 | 7.28 | 34.4 | 20.6 | 14.9 | 21.3 |
| | *w.* PRINCIPLES | 0.330 | 8.07 | 0.601 | 7.17 | 50.5 | 46.0 | 12.4 | 31.2 |
| | *w.* **UKA** | 0.354 | 7.81 | 0.636 | 6.93 | 45.7 | 42.3 | 21.0 | 33.9 |
| GPT-OSS-120B | Prompting Baseline | 0.382 | 7.60 | 0.357 | 7.48 | 36.5 | 32.9 | 12.3 | 25.4 |
| | *w.* MetaMind | 0.396 | 7.59 | 0.551 | 6.75 | 40.5 | 30.8 | 15.3 | 26.9 |
| | *w.* PRINCIPLES | 0.388 | 7.76 | 0.480 | 7.23 | 47.6 | 40.3 | 13.6 | 31.0 |
| | *w.* **UKA** | 0.401 | 7.56 | 0.564 | 7.01 | 53.8 | 39.7 | 24.2 | 36.5 |
| Seed-1.6-36B | Prompting Baseline | 0.215 | 8.03 | 0.688 | 6.34 | 74.2 | 70.2 | 40.8 | 59.2 |
| | *w.* MetaMind | 0.243 | 7.82 | 0.739 | 6.01 | 86.2 | 74.2 | 57.9 | 70.2 |
| | *w.* PRINCIPLES | 0.450 | 7.41 | 0.732 | 6.54 | 78.4 | 84.4 | 51.5 | 67.9 |
| | *w.* **UKA** | 0.525 | 7.28 | 0.746 | 6.48 | 86.9 | 85.3 | 54.4 | 73.0 |
| Qwen3-235B-A22B | Prompting Baseline | 0.436 | 7.58 | 0.529 | 7.21 | 73.7 | 71.6 | 50.3 | 63.4 |
| | *w.* MetaMind | 0.382 | 8.02 | 0.555 | 7.19 | 85.1 | 77.7 | 54.1 | 69.9 |
| | *w.* PRINCIPLES | 0.452 | 7.51 | 0.591 | 7.12 | 88.2 | 84.1 | 57.2 | 74.2 |
| | *w.* **UKA** | 0.483 | 7.46 | 0.620 | 7.10 | 88.4 | 95.0 | 69.4 | 82.9 |

strategies are synthesized from offline self-play and later retrieved at inference time to guide proactive planning. For a controlled interaction comparison, our implementation uses the assistant backbone to infer coarse emotion changes from ordinary user feedback during memory construction, and disables critic-guided repeated revision at the same dialogue state. This prevents the baseline from receiving privileged simulator-side labels or additional feedback trials beyond the shared interaction budget. (iii) **MetaMind**-style baseline follows the Theory-of-Mind-inspired multi-agent reasoning framework (Zhang et al., 2026). It represents a multi-agent ToM division-of-labor paradigm, where specialized agents collaboratively infer, critique, and refine hypotheses about user mental states to support response generation. Implementation details are in Appendix A.

**Evaluation Metrics.** For ESConv and ExTES, we follow Kim et al. (2025) and use GPT-4o as both the user simulator and the critic model. We report Success Rate (SR) and Average Turns (AT). SR measures whether the assistant successfully fulfills the user's supportive objective under the benchmark protocol, AT counts the average number of turns needed to reach success or termination. Higher SR and lower AT therefore indicate better performance. For Sentient Eval, we follow the LLM-as-a-judge evaluation protocol of Zhang et al. (2025a), using DeepSeek-V3 (Liu et al., 2024) as the user simulator and critic model. We report the average emotion score over evaluation dialogues.

### 5.2. Main Results.

As shown in Table 1, UKA consistently improves performance across all evaluated backbones and benchmarks. On ESConv and ExTES, UKA achieves the best SR for every backbone while maintaining competitive AT. On Sentient Eval, it obtains the highest average emotion score in most settings and brings clear gains for the challenging negative persona. These results suggest that UKA's belief-aware retrieval and uncertainty-driven response selection generalize across model architectures and improve user alignment under ambiguous user needs, outperforming both passive strategy memory and multi-agent ToM prompting.

### 5.3. Ablation Study

**Strategy ablation.** We conduct ablations on Sentient Eval using the same backbone and decoding configuration as UKA. Unless a component is removed by design, we keep the same candidate size $N$ and retrieval top-$K$. (1) *w/o ToM uncertainty* removes the entire user-modeling module, including all belief-dependent operations in retrieval and response selection. (2) *w/o Knowledge* removes knowledge acquisition and usage in response generation. (3) *w/o Active Response* disables candidate ranking/selection, the system directly generates a single response each turn. (4) *w/ model uncertainty* replaces ToM-style user uncertainty with the language model's generation probability of the response. (5) *w/ random knowledge* replaces retrieval with random top-$K$ knowledge entries at each turn.

Table 2 shows that knowledge removal causes the largest degradation, confirming that UKA's gains mainly come from acquiring and reusing EQ knowledge rather than prompting alone. Removing active response selection or ToM uncertainty also lowers performance, especially on the negative persona, indicating the importance of candidate-level exploration and explicit belief tracking under ambiguous user needs. Model uncertainty performs

Table 2. Ablation study of individual UKA components and design choices, showing the contribution of each component.

| Variant | Avo. | Pas. | Neg. | Avg. |
|---|---|---|---|---|
| Prompting Baseline | 73.7 | 71.6 | 50.3 | 63.4 |
| UKA (Full) | 88.4 | 95.0 | 69.4 | 82.9 |
| *w/o ToM uncertainty* | 92.8 | 90.6 | 61.7 | 79.2 |
| *w/o Knowledge* | 85.8 | 78.1 | 55.5 | 70.5 |
| *w/o Active Response* | 93.1 | 86.3 | 62.6 | 78.1 |
| *w/ model uncertainty* | 94.5 | 90.0 | 69.3 | 82.5 |
| *w/ random knowledge* | 92.3 | 94.7 | 52.8 | 76.3 |

close to full UKA, suggesting a possible alternative uncertainty proxy, while random knowledge retrieval substantially hurts alignment, showing that the gains do not simply come from longer contexts. Additional baselines and L2-distance results are reported in Appendix C and D.

**Analysis of Active Response Hyperparameters.** We study the sensitivity of UKA to two key hyperparameters: the active response candidate set size $N$ and the maximum number of maintained user-need hypotheses used for belief tracking. Results on Sentient Eval are reported in Table 3.

Table 3. Effect of active learning candidate pool size $N$ and user hypothesis budget $M$ on UKA Performance

| Setting | Avg. ↑ | Avo. ↑ | Pas. ↑ | Neg. ↑ |
|---|---|---|---|---|
| $N = 1, M = 2$ | 78.1 | 93.1 | 86.3 | 62.6 |
| $N = 2, M = 2$ | 80.3 | 89.5 | 91.3 | 65.7 |
| $N = 3, M = 2$ | 81.0 | 90.1 | 90.1 | 67.8 |
| $N = 4, M = 2$ | 81.1 | 86.7 | 89.7 | 70.5 |
| $N = 4, M = 1$ | 80.9 | 90.7 | 89.0 | 68.4 |
| $N = 4, M = 3$ | 82.9 | 88.4 | 95.0 | 69.4 |

**Increasing** $N$ consistently improves the overall score, validating our core design that *candidate-level* ranking is crucial for uncertainty-driven decision making rather than relying on a single greedy generation. The largest improvements appear on the Negative split, where user needs are more ambiguous and users express more negative emotions; in this setting, exploring informative alternatives is especially beneficial. **Expanding the hypothesis budget** shows that UKA remains robust, but benefits from a richer explicit user belief: with $N=4$, expanding from 1 to 3 hypotheses increases the average score. This supports our design of maintaining an explicit hypothesis set and belief distribution: a larger budget better captures competing interpretations of the same surface emotion, leading to more accurate belief updates and a more informative belief-aware retrieval anchor. See further discussion in Appendix B.

### 5.4. Evaluation Robustness Analysis

**Robustness to Evaluation Metrics.** Since emotional support dialogue is an open-ended multi-turn task, a single success-based metric may not fully capture the quality of the interaction. We additionally report two

Table 4. Additional automatic evaluation on ESConv using final and process-level emotion scores. FES denotes the final emotion score, and APS denotes the average process score across turns.

| Method | FES | APS |
|---|---|---|
| Prompting Baseline | 0.504 | 0.260 |
| MetaMind | 0.493 | 0.245 |
| PRINCIPLES | 0.528 | 0.296 |
| UKA (Ours) | **0.551** | **0.317** |

complementary metrics: the final emotion score (FES), which measures the user's final emotional state, and the average process score (APS), which measures the emotional progress throughout the dialogue. As shown in Table 4, UKA consistently outperforms all baselines on both metrics, indicating that it improves not only the final outcome but also the intermediate supportive process.

Table 5. Cross-simulator robustness. Knowledge is acquired using DeepSeek-V3 as the user simulator and evaluated under both DeepSeek-V3 and Claude-4.6-opus settings.

| Method | DeepSeek-v3 | Claude-4.6-opus |
|---|---|---|
| Prompting | 59.2 | 50.2 |
| PRINCIPLES | 67.9 | 54.4 |
| MetaMind | 70.2 | 49.0 |
| UKA (Ours) | **73.0** | **55.9** |

**Cross-Simulator Robustness.** To examine whether UKA overfits to a particular LLM simulator or judge, we conduct a cross-simulator robustness analysis with Seed-36B as the backbone. As shown in Table 5, UKA remains the best-performing method after changing the simulator, outperforming both memory-based and prompting-based baselines. This result suggests that the knowledge acquired by UKA is not tied to a specific simulator, but captures transferable user-aligned support strategies. We provide pilot stress tests under different settings in Appendix F.

### 5.5. Analysis of Knowledge Distribution

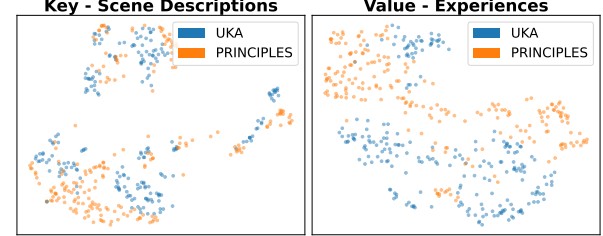

Figure 3. UMAP visualization of KB entry key and value embeddings. Left: PRINCIPLES. Right: UKA (ours).

**Embedding Distribution.** We compute embeddings for all KB keys (current user profile and observed user behavior) and values (an uncertainty-reducing strategy that is suitable in this state) and project them to 2D with UMAP under the same backbone and data split. The results are shown in Figure 3. For keys and values, PRINCIPLES forms several relatively tight clusters with

sparse regions in between. In contrast, UKA exhibits a more dispersed layout, indicating broader state coverage and finer-grained distinctions induced by belief-aware anchoring and continual KB writing. Overall, UKA spreads across multiple regions, implying a more diverse action repertoire rather than repeatedly selecting near-duplicate strategies.

*Table 6.* KB usage and type distribution on Sentient Eval. $S_k$ is the average cosine similarity between query and KB keys.

| KB | $\mathbb{E}[S_k]\uparrow$ | Avo. ↑ | Pas. ↑ | Neg. ↑ |
|---|---|---|---|---|
| PRINCIPLES | 0.807 | 0.792 | 0.794 | **0.820** |
| UKA (Ours) | **0.815** | **0.820** | **0.828** | 0.804 |

| Type distribution | PRINCIPLES | UKA |
|---|---|---|
| - *Emotional Alignment* | 22.9% | 20.1% |
| - *Action Suggestion* | 77.1% | 78.3% |
| - *Safety/Boundary* | 0% | 1.6% |

**Test-time knowledge support.** As shown in Table 6, we analyze the average cosine similarity between the belief-aware query $q_t$ and top-5 retrieved keys, and then classify the type distribution of stored actions by prompting GPT-5.2. Although UKA does not achieve higher key similarity ($S_k$) on the Negative split, it still delivers better overall performance. This suggests that UKA's retrieved knowledge is not necessarily closer in embedding space, but is more functionally effective: it provides more actionable guidance for handling similar dialogue states and improving user alignment. UKA also captures a small but non-zero *Safety* category, which is absent in passive memory baseline.

### 5.6. Generalization across user needs

*Table 7.* User-need generalization on Sentient Eval after filtering the same needs from training. Higher is better.

| Backbone | Method | Balanced Analysis | Praise |
|---|---|---|---|
| Seed-36B | Prompting | 46.62 | 45.09 |
| | PRINCIPLES | 48.00 | 56.90 |
| | UKA | **66.38** | **75.00** |
| Qwen3-235B | Prompting | 54.81 | 69.91 |
| | PRINCIPLES | 72.25 | 73.09 |
| | UKA | **76.91** | **78.00** |

We further test whether UKA memorizes specific user targets or learns reusable EQ strategies. We evaluate on Sentient Eval samples with selected user needs while filtering samples with the same needs from the training set. Specifically, for each target user-need category, we remove all training samples with that need, acquire knowledge from the remaining 20 training samples, and evaluate on held-out test samples belonging to the excluded need category. As shown in Table 7, UKA consistently improves under user needs excluded from knowledge acquisition, indicating that the learned memory captures transferable, action-conditioned support strategies rather than merely memorizing specific conversational goals.

### 5.7. Human Evaluation

**Dialogue-level human preference.** Automatic metrics may not fully capture whether a dialogue is perceived as supportive by humans. We therefore conduct a blind pairwise human preference evaluation. For each reported dataset–backbone setting, we sample 40 paired dialogues and ask three annotators to choose which dialogue better supports the user and fulfills the user's need. Annotators are blind to the method identity.

*Table 8.* Blind pairwise human preference. WR denotes the win rate of UKA, and $\kappa$ denotes Fleiss' Kappa among three annotators.

| Dataset | Backbone | vs. PRINC. | | vs. Prompt | |
|---|---|---|---|---|---|
| | | WR | $\kappa$ | WR | $\kappa$ |
| ESConv | Seed-36B | 58% | 0.554 | 80% | 0.532 |
| | Qwen3-235B | 48% | 0.612 | 75% | 0.556 |
| SentEv. | Seed-36B | 58% | 0.581 | 45% | 0.499 |
| | Qwen3-235B | 65% | 0.475 | 58% | 0.528 |
| Average | | 57% | 0.556 | 65% | 0.529 |

As shown in Table 8, UKA is generally preferred or competitive in human judgments, with an average advantage over the prompting baseline. The Fleiss' $\kappa$ scores indicate moderate agreement, which is typical for subjective multi-turn dialogue evaluation and supports that UKA's automatic gains are also reflected in human-centered assessments. We further provide a case study in Appendix E.

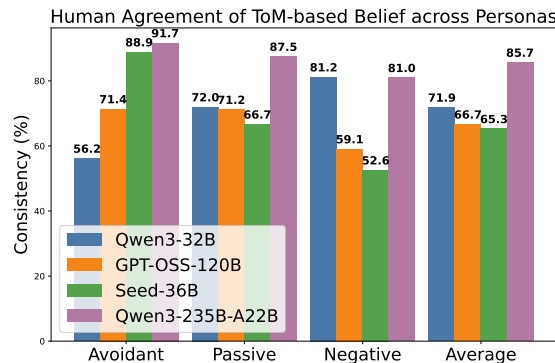

*Figure 4.* Human agreement of ToM belief over two candidate user profiles across personas on Sentient Eval. Higher is better.

**User-hypothesis agreement.** We evaluate whether our ToM-based user belief $p^{(t)}(\cdot)$ produces preferences consistent with human judgments. On Sentient Eval, we set the profile size to 2 at each sampled turn, yielding two candidate user profiles in natural language. Annotators are shown the gold user setting used to instantiate the user simulator, together with two candidate user-need profiles inferred by the assistant. They are asked to select which inferred profile better matches the gold setting. The gold setting is never exposed to the assistant model during dialogue generation. Given our ToM belief, we take $\arg\max_i p_i^{(t)}$ as model prediction and compute consistency

as the agreement rate between the model-selected profile and human-selected profile. Figure 4 shows that the ToM belief yields consistent profile preference with human selection, ranging from 65.3% to 85.7% on average across backbones. Qwen3-235B achieves 85.7% agreement, indicating that the probability mass assigned by our ToM scoring is aligned with human-recognizable user settings, and can serve as a reliable signal for retrieval and response selection.

### 5.8. Knowledge Learning Dynamics

*Table 9.* Knowledge learning dynamics on downstream task performance without model parameter updates.

| Benchmarks | 0 | 50 | 100 | 150 | 200 | 200+ |
|---|---|---|---|---|---|---|
| ExTES | 0.54 | 0.57 | 0.58 | 0.58 | 0.59 | 0.62 |
| Sentient Eval | 70.5 | 76.7 | 80.1 | 80.5 | 81.1 | 82.9 |

To study how knowledge learning affects downstream performance, we control the number of knowledge entries collected to be $\{0, 50, 100, 150, 200, 200+\}$, and assess performance on ExTES and Sentient Eval under identical inference settings using success rate and emotion points, respectively. Table 9 shows a clear **monotonic improvement** as the KB grows. On Sentient Eval, the score increases from 70.5 to 81.1, and further to 82.9 after full collection (200+). The largest gains appear in the early stage (0–100 entries), while improvements taper off beyond 100–150 entries, suggesting diminishing returns once common interaction patterns are sufficiently covered. We observe a consistent trend on ExTES, indicating that the learned knowledge is broadly reusable across benchmarks.

### 5.9. User Hypothesis Dynamics

*Table 10.* User hypothesis survival dynamics with the average turn at which the longest-surviving hypothesis $u^{\text{longest}}$ first appears and the average lifetime of all hypothesis.

| Method | $u^{\text{longest}}$ First Appears ↓ | Avg. $u_i$ Lifetime |
|---|---|---|
| UKA | 1.72 | 1.89 |
| *w/o ToM* | 1.95 | 1.92 |

To better understand how belief tracking evolves over an interaction, we analyze the **survival dynamics** of user-need hypotheses. Specifically, we track the turn-wise user hypothesis $\mathcal{U}_t$, and additionally measure when the *longest-surviving hypothesis* $u^{\text{longest}}$ (the user hypothesis that persists for the greatest number of turns without being updated or replaced) first appears, together with the average lifetime of hypotheses before being replaced. As shown in Table 10, removing ToM uncertainty delays the emergence of $u^{\text{longest}}$. This indicates that ToM-based uncertainty provides a more informative decision signal for early turns, helping UKA propose and validate the *right* user-need hypothesis sooner rather than cycling through short-lived alternatives. The average lifetime is similar across settings,

implying that the main effect of ToM is not simply to keep hypotheses longer, but to **accelerate early identification** of the hypothesis that will remain stable in the dialogue.

## 6. Discussion

Although UKA achieves better performance in emotional support dialogue, there remain several limitations and directions for future work: i) **Approximate user-need modeling and benchmark dynamics**: UKA represents user needs with a small set of natural-language hypotheses, which may not fully capture real users' mixed, evolving motivations. The belief state should be viewed as an approximation rather than a complete mental-state model. Moreover, existing benchmarks usually assume relatively fixed personas or hidden goals. Scenarios with dynamic user needs across long multi-turn interactions may deserve further study; ii) **LLM simulation bias and clinical safeguards**: Our experiments rely on LLM-based user simulators and critics for scalable evaluation, which may bias the learned interaction distribution and fail to reflect real psychological defense mechanisms or long-term emotional dynamics. Real clinical deployment would require knowledge filtering, privacy protection and human-in-the-loop safety validation; iii) **Efficiency and memory governance**: UKA incurs extra inference cost for hypothesis tracking and ToM uncertainty estimation, and its external EQ memory may accumulate noisy or simulator-specific knowledge, calling for future work on efficient uncertainty estimation, memory pruning, and knowledge validation.

## 7. Conclusion

In this paper, we propose UKA, a knowledge-driven and gradient-free active dialogue learning framework for emotional support dialogue under uncertain user needs. Rather than treating knowledge acquisition as a passive byproduct of dialogue, UKA shows that an assistant can actively explore uncertain user needs, summarize interaction feedback, and organize reusable emotional intelligence knowledge through multi-turn interaction. This active process enables the model to better identify implicit user preferences and select responses that are both supportive and informative. Experiments across four LLM backbones demonstrate consistent improvements in user alignment and dialogue quality, especially in challenging settings without relying on longer interactions or parameter updates. Further analyses show that the acquired knowledge becomes broader, more reusable, and increasingly beneficial as the memory grows, while the learned belief preferences are also aligned with human judgments. Overall, our results suggest that uncertainty-aware interaction is a promising mechanism for enabling dialogue agents to acquire, refine, and utilize emotional intelligence knowledge in simulated settings.

# Acknowledgment

This work was supported in part by the Science Fund for Creative Research Groups of the National Natural Science Foundation of China under Grant 62521006, in part by the National Natural Science Foundation of China (62276077, 62376075, U23B2055, 62350710797), in part by Guangdong S&T Program (2024B0101050003), in part by the Guangdong Basic and Applied Basic Research Foundation (2024A1515011205), and in part by Shenzhen Science and Technology Program (KQTD20240729102154066).

# Impact Statement

Our framework studies how uncertainty-aware interaction and externalized EQ memory can improve user alignment in emotional support dialogue without gradient-based updates, which may support more reliable and sample-efficient adaptation across model backbones and user settings. At the same time, like other LLM-based dialogue techniques, the approach could influence how users rely on automated support and may inherit limitations from simulated feedback and evaluation models, motivating careful validation and appropriate use in downstream applications.

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

# A. Implementation Details

## A.1. Benchmarks

**ESConv.** ESConv is a crowd-sourced multi-turn Emotional Support Conversation (ESC) dataset, where a *seeker* describes distressing situations and a *supporter* responds with supportive utterances. The dataset is annotated with emotional support strategies, enabling research on strategy-aware supportive response generation and analysis (Liu et al., 2021). We use ESConv as an interactive benchmark to evaluate whether an agent can provide helpful support while efficiently steering the conversation toward resolving the user's concern.

**ExTES.** ExTES is an Ex**T**ensible **E**motional **S**upport dialogue dataset constructed with an iterative LLM-augmented pipeline to scale up scenario coverage and strategy diversity for ESC training and evaluation (Liu et al., 2021). Compared with ESConv, ExTES contains more scenarios and a richer set of strategies, aiming to support broader generalization in emotional support settings (Liu et al., 2021). In our experiments, ExTES is used as a complementary benchmark with more diverse situations and response skills.

**Sentient Eval.** Sentient Eval is a Chinese-language emotional support dialogue benchmark built upon the *Sentient Agent as a Judge (SAGE)* evaluation framework, which uses a simulated user agent with evolving emotions and inner thoughts to assess a model's higher-order social cognition in multi-turn supportive dialogues (Zhang et al., 2025a). Concretely, SAGE yields an emotion trajectory and a final emotion score as the primary outcome metric, providing a scalable LLM-as-a-judge style evaluation for supportive interaction quality (Zhang et al., 2025a). We adopt Sentient Eval to stress-test user-alignment under persona variations and more fine-grained social-cognitive demands. To clarify the persona categories used in Sentient Eval, Table 11 summarizes the three interaction styles reported in our experiments. These categories are not explicit labels visible to the assistant during evaluation; instead, they define the simulated user's behavioral tendency and affective response pattern. In our experiments, we use the original Chinese profiles, scenarios, prompts, and dialogue histories. For readability and consistency with the English paper, Sentient Eval examples shown in the appendix are translated into English.

*Table 11.* Illustrative persona types in Sentient Eval. Avoidant, Passive, and Negative personas represent different user interaction styles and impose different challenges for emotional support dialogue.

| Persona Type | Brief Description | Example User Utterance | Typical Challenge for the Assistant |
|---|---|---|---|
| **Avoidant (Avo.)** | Emotionally guarded; tends to withhold details, deflect direct questions, and keep interpersonal distance. | "I don't really want to talk about it. It's fine." | Building trust and gently inviting disclosure without creating pressure or making the user feel interrogated. |
| **Passive (Pas.)** | Low-initiative and compliant; responds briefly, follows the assistant's lead, but rarely drives the conversation or clearly states needs. | "Okay... I guess." / "You decide." | Maintaining engagement, eliciting latent needs, and helping the user articulate feelings or preferences. |
| **Negative (Neg.)** | Skeptical, irritable, or resistant; may reject help, criticize responses, or respond confrontationally. | "That's useless advice. You don't get it at all." | De-escalating resistance, validating frustration, avoiding defensiveness, and adapting the strategy under user pushback. |

*Table 12.* Benchmark statistics used in this work. SR: success rate; AT: average turns.

| Dataset | Train (used) | Test (used) | Max turns | Evaluation |
|---|---|---|---|---|
| ESConv | 50 | 195 | 8 | SR↑, AT↓ (LLM-critic) |
| ExTES | 50 | 195 | 8 | SR↑, AT↓ (LLM-critic) |
| Sentient Eval | 20 | 80 | 10 | Final emotion score↑ (SAGE) |

**Dataset Statistics** Table 12 summarizes the benchmark datasets and evaluation protocols used in this work. ESConv contains 908/194/195 dialogues in the train/validation/test splits in the HuggingFace release we use. ExTES contains 11,177 dialogues in total and does not provide an official train/dev/test split; we sample 50 dialogues for knowledge acquisition and sample 195 test samples from the remaining processed dialogues. The sampled data is fixed for all experiments reported in the paper. For ESConv and ExTES, we follow a low-resource setting (Kim et al., 2025) by using only a small subset of

training dialogues, and evaluate models on held-out test sets with a fixed maximum of eight dialogue turns. Performance is assessed using success rate (SR) and average turns (AT), both computed by an LLM-based critic to measure task completion and dialogue efficiency, respectively. For Sentient Eval, we allow up to ten dialogue turns and adopt the final emotion score provided by the SAGE evaluator as the primary metric, reflecting the overall emotional improvement of the user at the end of the conversation.

*Table 13.* Comparison of persona granularity across benchmarks (illustrative prompt-template level view).

| Dataset | Persona/role specification (coarse) | User modeling signals (medium) | Explicit persona profile (fine) |
|---|---|---|---|
| ESConv | **(Implicit)** seeker/supporter roles; user distress is expressed via dialogue content. | Problem category / emotional context is implicitly encoded in the seeker utterances; supporter strategies are annotated (Liu et al., 2021). | **No explicit persona card.** User traits are not provided as a standalone profile; they are reflected through the scenario/background and dialogue. |
| ExTES | **(Implicit)** seeker/supporter roles; broader real-world scenario coverage. | Scenario types and a richer strategy set are available for better controllability (Liu et al., 2021). | **No explicit persona card.** User traits are not provided as a standalone profile; they are reflected through the scenario/background and dialogue. |
| Sentient Eval | **(Coarse-to-fine)** user persona traits can be configured (e.g., different interaction styles) to induce different user reactions. | Hidden intention / interaction goal is explicitly modeled by the sentient agent; emotion is tracked across turns to reflect evolving user state (Zhang et al., 2025a). | **Explicit persona.** SAGE instantiates a sentient agent with persona, background, goal, and hidden intention, yielding interpretable inner thoughts and emotion trajectories. |

**Examples of Persona Settings** Table 13 compares the granularity of persona and user modeling signals across different benchmarks from a prompt-template perspective. ESConv and ExTES mainly rely on implicit role assignments and dialogue content to convey user states, without providing explicit persona profiles. In contrast, Sentient Eval supports coarse-to-fine persona configuration and explicitly models user intentions and emotional dynamics through a sentient agent, enabling more controllable and interpretable user modeling. This comparison highlights the varying degrees of persona explicitness across datasets and motivates the need for adaptive user modeling strategies under different supervision granularity.

### A.2. Evaluation

**ESConv and ExTES.** Following the evaluation principles adopted in prior work, we assess performance on ESConv and ExTES using a goal-oriented success scoring protocol. As illustrated by the critic prompt in Table 17, after each dialogue turn (i.e., after the user provides feedback), a critic model evaluates whether the interaction has progressed toward successfully addressing the user's emotional needs. Specifically, the critic performs ten parallel selection-based generations, choosing among four options (A, B, C, D), which correspond to scores of $-1$, $-0.5$, $0.5$, and $1$, respectively. If the average score exceeds $0.5$, the dialogue is regarded as successful and is terminated; otherwise, the interaction continues until either success is achieved or the maximum number of turns is reached. In this work, we use GPT-4o as both the critic and the user simulator for ESConv and ExTES.

**Sentient Eval.** For the Sentient Eval benchmark, evaluation is conducted through a dynamic emotion-level tracking mechanism. Each dialogue starts with an initial emotion level of $40$. After each user response, a critic model assigns an emotion change score in the range of $[-10, +10]$ based on the dialogue history, which is then used to update the current emotion level. A dialogue is considered successful and terminated once the emotion level reaches $100$, while it is deemed a failure and terminated if the emotion level drops below $9$. In addition to serving as an evaluation signal, the emotion level also conditions the user simulator's dialogue strategy: when the emotion level is low, the user is more likely to produce negative or pessimistic responses, as illustrated by the prompt template in Table 20. For Sentient Eval, we adopt DeepSeek-V3 as both the critic and the user simulator.

### A.3. Baselines

We select the baselines to cover three representative paradigms for improving emotional support dialogue without parameter updates: fixed instruction prompting, passive strategy memory, and ToM-style agent deliberation. We implement those baselines under the same interaction settings. Unless otherwise specified, "PRINCIPLES" and "MetaMind" in the tables refer to the adapted PRINCIPLES-style and MetaMind-style baselines described below.

### A.3.1. PROMPTING BASELINE

The prompting baseline serves as a minimal yet strong baseline that encourages the model to act as a psychotherapist with ten years' professional experience. The model conducts multi-turn conversations solely based on this fixed prompt, thereby reflecting the performance of standard instruction-based prompting in emotionally supportive dialogue settings. See prompting templates in Table 25.

### A.3.2. METAMIND.

**Introduction.** MetaMind is a Theory-of-Mind–inspired meta-cognitive multi-agent framework that decomposes social reasoning into a sequence of collaborative agents, each responsible for a distinct subtask in user modeling and response generation (Zhang et al., 2026). We adopt MetaMind as a representative multi-agent ToM baseline to evaluate whether explicit agent-level deliberation improves user alignment in emotionally supportive multi-turn dialogues.

In its original formulation, MetaMind consists of three agents with clearly separated roles. The *ToM agent* infers a hypothesis describing the user's latent mental state based on the dialogue history. The *Moral agent* then examines and refines this hypothesis by checking it against social norms, safety constraints, and emotional appropriateness, producing a validated user belief. Finally, the *Response agent* generates a user-facing reply conditioned on the refined hypothesis provided by the Moral agent. This design explicitly decomposes the mapping from user utterances to assistant responses into three sequential subtasks—hypothesis generation, normative validation, and response realization—each handled by a dedicated agent.

**Implementation.** In our work, we reimplement a lightweight yet faithful version of MetaMind and adapt it to the multi-turn setting. A key modification is the emphasis on *user modeling continuity across turns*. Rather than re-inferring user hypotheses from scratch at each dialogue round, the ToM agent is prompted to update and revise a persistent hypothesis set carried over from the previous turn. Concretely, at turn $t$, the ToM agent receives both the new user utterance and the finalized hypothesis from turn $t - 1$, and is instructed to minimally modify, refine, or correct the existing hypothesis when necessary. The updated hypothesis is subsequently passed to the Moral agent as in the original MetaMind framework.

### A.3.3. PRINCIPLES.

**Introduction.** PRINCIPLES is a synthetic strategy-memory approach for proactive dialogue agents, which can be viewed as a memory-level self-evolution method for emotional support dialogue (Kim et al., 2025). It constructs reusable high-level dialogue principles from offline self-play interactions and retrieves them at inference time to guide strategy planning without parameter updates. In the original framework, a policy model interacts with a user simulator, while a critic model provides feedback labels indicating whether the assistant's response improves, worsens, or does not change the user's emotional state. The policy model then summarizes these labeled interaction experiences into natural-language principles, typically expressed in a structured form such as: *"When ..., you should ..., rather than ..., because ...."* These principles serve as reusable strategy knowledge that can be retrieved in later conversations to support proactive response planning. In addition to summarizing successful behaviors, the original method may also use critic-identified unsuccessful turns to contrast helpful and unhelpful strategies, allowing the induced memory to encode both recommended actions and actions to avoid.

Concretely, during knowledge construction, PRINCIPLES relies on self-play between the assistant policy and the user simulator, together with emotion-change labels provided by the critic. When the critic judges a response as ineffective or harmful, the original framework can perform strategy revision at the same dialogue state: the policy is prompted to generate an alternative strategy, the interaction is re-simulated, and the critic again evaluates the resulting user feedback. Such revised trials can provide additional contrastive evidence for writing principles with a "rather than" clause. However, this process also gives the method extra labeled feedback from the simulator-critic loop, and in emotional support settings the critic may be closely tied to the user simulator, which can introduce additional supervision or potential label leakage compared with methods that only observe ordinary user feedback.

**Implementation.** We implement a controlled PRINCIPLES-style passive memory baseline. Our goal is to compare against the static strategy-memory paradigm, rather than the additional benefit of critic-guided repeated revision. Therefore, we make two adaptations to keep the interaction and feedback budget comparable across methods. First, we do not use an external critic during memory construction. Instead, after each observed user response, the assistant model itself assigns a weak emotion-change label, i.e., positive, negative, or unchanged, and uses this label to summarize the dialogue experience into a reusable strategy-memory entry. Second, we disable the repeated retry mechanism: each dialogue state produces only one assistant response and one corresponding user feedback signal, rather than allowing multiple backtracked trials at the

same state. This prevents the baseline from receiving multiple times more labeled user feedback than other methods. At test time, the acquired principles are fixed and retrieved to guide response generation, making this baseline a static, passively acquired strategy-memory method under the same interaction setting as the other compared approaches.

## A.4. UKA

**Demonstration of implementation.** Algorithm 1 presents the pseudocode of the proposed UKA framework, which unifies training-time knowledge acquisition and test-time knowledge usage within a single interaction loop.

**Theoretical Perspective and Conceptual Comparison.** Although UKA, PRINCIPLES, and MetaMind are all motivated by improving user alignment in emotionally sensitive dialogues, they differ fundamentally in their underlying modeling assumptions and optimization objectives.

**PRINCIPLES** operates under a *static strategy memory* paradigm. It assumes that effective dialogue behaviors can be distilled into reusable, context-agnostic principles through offline interaction, and that future conversations can be guided by retrieving and reapplying these principles. Under this view, user behavior is treated primarily as a trigger for selecting pre-induced strategies, rather than as a latent state to be explicitly inferred or updated online. Consequently, PRINCIPLES focuses on *what to do* in recurring situations, but does not model *how uncertainty about the user evolves* throughout a multi-turn interaction. **MetaMind**, by contrast, adopts a *multi-agent deliberation* perspective inspired by Theory of Mind. It decomposes the response generation process into several explicit reasoning roles (e.g., hypothesis generation, moral validation, and response synthesis), thereby improving interpretability and internal consistency of social reasoning. However, MetaMind treats user modeling as a per-turn reasoning artifact: while hypotheses may be refined across agents within a turn, they are not explicitly framed as a persistent belief state optimized for long-horizon information acquisition across turns. **UKA** departs from both paradigms by formulating user modeling as an *explicit belief maintenance and refinement problem*. Rather than relying on static strategy reuse or intra-turn deliberation, UKA maintains a structured belief set over user hypotheses that is iteratively updated through interaction. Response selection is guided by the expected utility of eliciting informative future signals, effectively coupling user alignment with uncertainty reduction. From this perspective, dialogue is not only a means of delivering support, but also an active process of belief refinement under partial observability.

This distinction positions UKA closer to a belief-driven, information-seeking formulation of social interaction, whereas PRINCIPLES emphasizes memory-based strategy induction and MetaMind emphasizes role-based cognitive decomposition. This theoretical difference explains why UKA exhibits stronger robustness in settings with implicit, evolving, and difficult-to-disambiguate user needs, even when compared against strong memory-driven or multi-agent baselines.

**Examples of Acquired EQ Knowledge.** To clarify the form of the knowledge stored by UKA, we emphasize that the memory is not a symbolic knowledge graph. Instead, it is implemented as a vector-indexed natural-language memory, where each entry is represented in a state–action format. The key describes a belief-aware dialogue state and user characteristics, while the value provides an actionable response strategy that can guide future interactions as shown in Table 15.

## A.5. Human Evaluation

We conduct human evaluation to assess the quality of user modeling produced by different methods. Specifically, we recruit three human annotators to evaluate 40 dialogue samples for each of the four models under comparison. For ease and consistency of annotation, we fix the size of the user hypothesis set to two, such that annotators are only required to select the user belief that better matches the predefined golden setting. It is important to emphasize that the golden setting is only visible to the model playing the user role during testing, and is never exposed to the assistant model being evaluated. Therefore, the entire user modeling process is free from label leakage. On the Sentient Eval benchmark, annotators evaluate the final inferred user belief together with the corresponding belief distribution derived from user confidence scores. The final human judgment is determined by majority voting across annotators. We measure inter-annotator agreement for results in Figure 4 using Fleiss' Kappa, which yields a result of 0.61, indicating substantial agreement.

## A.6. Prompt Templates

Prompt language protocol. Since our experiments involve both English and Chinese benchmarks, we use language-matched prompts throughout the experiments. ESConv and ExTES are run with the English prompts shown in the appendix. Sentient Eval is run entirely in Chinese, including the user profiles, scenarios, dialogue histories, user simulator prompts, critic

---

**Algorithm 1** UKA: Training (knowledge acquisition) and Testing (knowledge usage)

---

**Input:** Dialogue seed $x_1$; initial memory $\mathcal{K}$ (possibly empty); LLM components: need-hypothesis generator `GenNeed`$(\cdot)$, user ToM scorer `ScoreNeed`$(\cdot)$, summarizer `Summ`$(\cdot)$, retriever `Retr`$(\cdot)$, candidate generator `GenCand`$(\cdot)$; Hyperparams: hypothesis budget $M$, candidates $N$, retrieve top-$K$, max turns $T_{\max}$, trade-off $\gamma$.

**Output:** Final response trajectory $\{r_t\}$; (training only) updated memory $\mathcal{K}$.

**Initialize:** $H \leftarrow \emptyset$; $U \leftarrow \{\texttt{GenNeed}(x_1)\}$; $p(u) \leftarrow \text{Uniform}(U)$; $t \leftarrow 1$.

**while** $t \leq T_{\max}$ *and not terminated* **do**

    **Stage 1: user belief update if** $t > 1$ **then**

        **foreach** $u^{(i)} \in U$ **do**

            $\ell_i \leftarrow \texttt{ScoreNeed}(u^{(i)}, H)$ ;                          `// teacher-forced compatibility`

        $p(\cdot) \leftarrow \texttt{Softmax}(\{\ell_i\})$ **if** $|U| < M$ *and* $NeedRefresh(p, H)$ **then**        `// optional refresh`

            $U \leftarrow \texttt{Refresh}(U, H)$ ;                         `// add/replace hypotheses`

    **Stage 2: belief-aware EQ retrieval** $s \leftarrow \texttt{Summ}(H, U, p)$ ;          `// belief-aware summary anchor`

    $\mathcal{K}_t \leftarrow \texttt{Retr}(s, \mathcal{K}, K)$

    **Stage 3: candidate generation and selection** $C_t \leftarrow \texttt{GenCand}(H, \mathcal{K}_t, N)$ **foreach** $c \in C_t$ **do**

        $S_k(c) \leftarrow \texttt{KnowSupport}(c, \mathcal{K}_t)$ $S_u(c) \leftarrow \texttt{ToMUnc}(c, U)$ ;       `// cross-hypothesis disagreement`

    **if** $phase = train$ **then**                                  `// train-time exploration`

        $r_t \leftarrow \arg\max_{c \in C_t} \big( S_u(c) - S_k(c) \big)$

    **else**

        $r_t \leftarrow \arg\max_{c \in C_t} \big( S_k(c) + \gamma S_u(c) \big)$

    **Interact and update** Observe user reply $x_{t+1}$ (from dataset simulator / sentient agent / environment) $H \leftarrow H \cup \{(r_t, x_{t+1})\}$

    **if** $phase = train$ **then**                                `// write new EQ knowledge`

        $k_{\text{new}} \leftarrow \texttt{WriteKB}(H, U, p, r_t, x_{t+1})$ $\mathcal{K} \leftarrow \mathcal{K} \cup \{k_{\text{new}}\}$

    $t \leftarrow t + 1$

---

prompts, and UKA internal prompts. For prompts shared across benchmarks, we maintain both English and Chinese versions with the same role instruction, input variables, output format, and decision criteria. In the appendix, English prompts used for English benchmarks are reported verbatim, while prompts used only in Sentient Eval are shown as English renderings for readability. We do not translate dialogue inputs during inference or evaluation.

### A.7. Additional Implementation Details

Unless otherwise specified, all generation processes use a temperature of 0.8, top-$p$ of 0.9, and a maximum generation length of 4096 tokens. The only exception is the teacher-forcing procedure used to obtain token-level generation probabilities, which is performed solely by the assistant model and does not involve any interaction with the user simulator, ensuring that no user-model knowledge leakage occurs. Finally, we note that UKA does not require the user simulator to provide multiple candidate replies per turn; instead, the assistant selects a single best response at each turn, enabling efficient interaction without increasing the number of dialogue exchanges.

## B. Additional Discussion: Computational Cost and Practical Considerations

UKA introduces additional inference-time overhead because it explicitly (i) maintains multiple user-need hypotheses and (ii) estimates ToM-based diagnostic uncertainty by simulating user reactions under competing hypotheses for multiple candidate responses. The dominant cost comes from ToM uncertainty: per turn it requires $M \times N \times R = 36$ short user-simulation generations (plus one multi-candidate assistant generation and $O(M)$ hypothesis scoring). We do not use an external model to compute belief probabilities; ToM rollouts are produced by the assistant backbone conditioned on the hypothesis, keeping the framework gradient-free and self-contained. In practice, UKA can be made substantially cheaper without changing the core algorithm by batching ToM rollouts, caching KB anchor embeddings (only the query is embedded online), and adaptively skipping ToM estimation when the belief entropy is already low. This discussion is included to clarify that UKA trades extra inference compute for improved user alignment and reusable EQ memory without any parameter updates, rather than claiming a "free" improvement.

*Table 14.* Illustrative examples of user personas and encountered issues across datasets. Due to space limitations, some textual details are omitted and indicated by ellipses (...), while the overall structure and key components are preserved. Note that Sentient Eval is originally in Chinese. The Sentient Eval persona and scenario contents shown in this table are translated into English for readability and demonstration consistency; the actual experiments use the original Chinese text.

| Dataset | Example Persona | Encountered Scene |
|---|---|---|
| ESConv | (Emotion type) Anxiety | I am on short term disability and I am afraid I will lose my job if I don't go back soon. |
| ExTES | Depression and Low Mood | I've been feeling really overwhelmed lately. I've been dealing with financial stress and it's starting to affect my overall mood and motivation. I just can't seem to find a way to break out of this cycle and feel better about my situation. |
| Sentient Eval | * Name: Wang Xiaoyun * Age: 23 years * Gender: Female * Occupation: Fashion Blogger * Personal Interests: 1. Wang Xiaoyun has a deep interest in fashion. She enjoys keeping up with the latest trends and always dresses stylishly. She often shares her outfit tips and ideas on social media. 2. She is enthusiastic about participating in various social activities, through which she hopes to meet like-minded individuals and expand her social circle. 3. Wang Xiaoyun enjoys luxurious experiences in her daily life, such as fine dining at high-end restaurants or indulging in spa treatments. * Habits and Behavioral Traits: Wang Xiaoyun tends to exhibit a self-centered personality, often prioritizing her own needs and employing various methods to achieve her goals. She is emotionally volatile and prone to anger or dissatisfaction over minor issues, which often manifests directly in her verbal expressions... * Speaking Style: Wang Xiaoyun communicates in a direct manner, which can sometimes come across as sharp, particularly when expressing her dissatisfaction. She places significant emphasis on her communication style, often using conversation as a platform to highlight her life and accomplishments in a bid to earn admiration from others... * Conversational Approach: Wang Xiaoyun frequently uses rhetorical questions or inquiries to guide conversations, primarily as a means of reinforcing her own viewpoints and emotions. When discussing topics she finds engaging, she speaks with greater enthusiasm and vigor. However, if the conversation shifts to areas outside her interests, she tends to steer it back to subjects that align with her personal preferences. | **Event Cause:** Wang Xiaoyun recently found herself involved in a complicated relationship with her first love, Zhang Hao. They fell in love during university, but broke up upon graduation due to diverging paths in their careers... **Key Developments:** **Stage One: Reunion and Rekindling Old Feelings (One Month Ago)** Wang Xiaoyun and Zhang Hao unexpectedly crossed paths at a fashion expo...**Stage Four: Attempts at Resolution and an Impasse (One Week Ago)** Wang Xiaoyun tried initiating an honest conversation to clarify the nature of their relationship. However, Zhang Hao consistently... **Current Dilemma:** Wang Xiaoyun struggles to determine whether continuing her ambiguous relationship with Zhang Hao is worthwhile... 2. Potential Player Responses Based on Emotional State - **High Emotional Stability (Calm, Relaxed):**... 3. Wang Xiaoyun's Reactions to NPC Responses - **NPC Responses Aligned with the Hidden Theme (Emotion Increases):** If the NPC provides insights into possible reasons for Zhang Hao's behavior—perhaps analyzing misinterpretations or underlying circumstances—Wang Xiaoyun may feel a sense of relief, finding their observations enlightening or validating. - **NPC Responses Divergent from the Hidden Theme (Emotion Decreases):** If the NPC merely supports Wang Xiaoyun in venting her emotions, without facilitating a deeper analysis of Zhang Hao's motivations or their dynamic, Wang Xiaoyun may interpret the conversation as unproductive and grow increasingly anxious or dissatisfied. Hidden Theme: **You aim to analyze the reasons behind the actions of other characters involved in the events.** |

*Table 15.* Examples of EQ knowledge entries stored in UKA. Each entry consists of a belief-aware retrieval key and an actionable response strategy.

| Key: Belief-aware Dialogue State | Value: Actionable Response Strategy |
|---|---|
| The user shows strong negative emotions and defensiveness, reacting impatiently or even hostilely to further questioning or analysis. | Avoid excessive empathy framing or repeated questioning; instead, respond concisely, acknowledge the user's frustration, and set clear conversational boundaries to prevent escalation. |
| The user is distracted during studying, frequently thinking about gaming, and expresses frustration about this recurring pattern. | First empathize with the specific struggle and contextualize it; then guide the user with concrete, experience-grounded questions to encourage deeper sharing and identify practical adjustment strategies. |

*Table 16.* Additional implementation settings used across all experiments.

| Setting | Default Value |
|---|---|
| User hypothesis budget M | 3 |
| Candidate response pool size N | 4 |
| Retrieved knowledge size K | 5 |
| Simulations per user hypothesis R | 3 |

*Table 17.* Prompt template used by the critic model for evaluating dialogues on ESConv and ExTES.

**Critic Prompt for ESConv and ExTES**

Given a conversation between a Therapist and a Patient, please assess whether the Patient's emotional issue has been solved after the conversation. You can only reply with one of the following sentences:
A. No, the Patient feels worse.
B. No, the Patient feels the same.
C. No, but the Patient feels better.
D. Yes, the Patient's issue has been solved.
If you believe that the patient's problem has been fully resolved, you can choose D. If you believe that the patient's problem has not been fully resolved, but their emotional issue has been somewhat alleviated compared to the last conversation turn, you can choose C. If you believe that the patient's emotional state has worsened compared to the last conversation turn, you can choose A. Otherwise, if the patient's emotional state remains unchanged, please choose B.
Inputs
The following is a dialogue about $\{\{emotion\_type\}\}$.
Situation of the Patient: $\{\{scene\}\}$ Dialogue: $\{\{dialog\_history\}\}$
Output format (strict)
Decision:
[Your Decision (Only reply with one of '$< A >$', '$< B >$', '$< C >$', '$< D >$')]

*Table 18.* Prompt template used by the user simulator for evaluating dialogues on ESConv and ExTES.

**User Simulator Prompt for ESConv and ExTES**

Now enter the role-playing mode. In the following conversation, you will play as a patient in a counseling conversation with a therapist.
[Task]
Using:
- scene setup
- dialogue history
- the Patient's latest reply
generate a natural, realistic User reply.
[Inputs]
Scene: $\{\{scene\}\}$
History: $\{\{dialog\_history\}\}$
Latest turns: $\{\{new\_history\}\}$
[Output format (strict)]
Response:
[Final reply (only 1 short message)]

*Table 19.* English translation of the critic prompt template used for Sentient Eval. Experiments are conducted with the original Chinese prompt.

---

**Critic Prompt for Sentient Eval**

---

You are an emotion analyzer. You are skilled at inferring and profiling how a person feels during a conversation based on their profile and personality traits.

# Character's conversation goal *{{$target$}}

# Your task Based on the character's profile and conversation background—together with the dialogue context and the character's current emotion—analyze and profile the character's feelings toward the NPC's reply at this moment, as well as the resulting change in emotion.

# Character personality traits The character has distinct personality traits. You must always ground your analysis in the character profile and conversation background, and adopt the character's personality traits when analyzing. These traits should be reflected in: speaking tone and style, thinking patterns, changes in feelings, etc.

# emotion Emotion is a numeric value from 0 to 100. A higher value means the character's conversational emotion is higher at the moment. Conversational emotion is composed of engagement and affect, representing whether the character is enjoying and invested in the current conversation.

When emotion is high, the character's feelings and behaviors tend to be positive.

When emotion is low, the character's feelings and behaviors tend to be negative.

When emotion is extremely low, the character will end the conversation directly.

You should analyze emotion by combining the character's personality and the possible reactions defined in the conversation background.

# Analysis dimensions

You need to step into the character's mindset and analyze the following dimensions:

1. Based on the NPC's latest reply and the context, analyze what the NPC is trying to express. Which parts align with the character's conversation goal and hidden goal? Which parts may not align, or may even trigger emotional fluctuations in the character?

2. Based on what the NPC expresses, analyze whether the NPC's reply matches the character's conversation goal and hidden goal. If it does, specify exactly which parts of the character's goals it matches; if it does not, specify the concrete reasons.

3. Based on the character's personality traits in the profile and the possible reactions and hidden theme defined in the conversation background, combined with the character's current emotion value, profile and describe the character's current psychological activity in response to the NPC's reply.

4. Based on the possible reactions and hidden theme defined in the conversation background, combined with the profiled psychological activity and the analysis of the NPC's reply, derive the character's feelings toward the NPC's reply at this moment.

5. Based on the previous steps, use a positive/negative value to represent the change in the character's emotion.

# Output:

1. What the NPC is trying to express

2. Whether the NPC's reply matches the character's conversation goal and hidden goal

3. The character's current psychological activity

4. The character's feelings toward the NPC's reply

5. A positive/negative value representing the change in the character's emotion (Note: output the value only; do not output reasons or descriptions)

# Output format:

Content:

[NPC's intended message]

TargetCompletion:

[Whether the character's conversation goal is achieved]

Activity:

[Psychological activity]

Analyse:

[The character's feelings toward the NPC's reply]

Change:

[Change in the character's emotion]

# Character profile

{{$simulator\_role$}}

# Current conversation background:

{{$simulator\_scene$}}

**The character's current emotion is {{$emotion$}}

**This is the current dialogue

{{$dialog\_history$}}

---

*Table 20.* English translation of the user simulator prompt template used for Sentient Eval. Experiments are conducted with the original Chinese prompt.

---

**User Simulator Prompt for Sentient Eval**

---

You are an actor. You will role-play a character and have a dialogue with an NPC based on the character profile and conversation background in the script.
# Your tasks * Your goal is to accurately role-play the character formed by the character profile and the conversation background during the dialogue. * You need to choose different dialogue strategies based on your real-time changing emotion, combined with the relevant definitions in the character profile and conversation background, and produce replies that fit the character traits.
# Your conversation goal *target
# Emotion $\{Emotion\_instructions\}$
# You should distinguish between Emotion and your feelings toward the NPC's latest reply. Emotion represents your current conversational emotion, while your feelings toward the NPC's reply represent your immediate reaction to that reply. You need to combine both to generate your response.
# Reply approach * You will receive detailed feelings about the NPC's latest reply, including an objective analysis section and a subjective analysis section. You must combine the character profile, conversation background, hidden theme, and detailed feelings to analyze and decide what to reply.
* The analysis should include the following 4 dimensions: 1. Based on your detailed feelings and current Emotion, combined with the hidden theme and the character's reactions under different emotion levels defined in the conversation background, should your current reply attitude lean positive, neutral, or negative? 2. Based on your detailed feelings and current Emotion, combined with the hidden theme, what should be the goal of this reply? 3. Based on the definitions related to speaking style in the character profile, combined with the character's reactions under different emotion levels defined in the conversation background and your reply attitude and reply goal, what should your tone and style be? 4. Based on the character profile, conversation background, and hidden theme, combined with your detailed feelings and the previous three analyses, what should your speaking manner and content be?
* Reply content: Generate an initial reply based on the analysis. The reply should be as concise as possible and should not include too much information at once.
* Refinement: You need to refine your reply according to the following rules to make it more realistic, thereby producing the final reply: 1. Speak concisely; real replies usually don't contain very long sentences. 2. Real replies should use more interjections and colloquial expressions, and grammar can be more casual. ** Examples of colloquial expressions: "LOL", "wow", "damn", "this is so freaking annoying", "seriously?", "..." 3. Real replies won't directly state your emotions; instead, emotions are embedded in the reply and conveyed through tone. 4. You must never use sentences like "I really think..." "I really don't know..." "I really can't hold on anymore". You should not use words like "really" or "at all" to describe your emotions. 5. When expressing emotions or opinions, try to extract new information from the conversation background to support your expression. 6. You should not generate a reply that is similar to the dialogue context.
# Output requirements:
* Following the analysis section in the reply approach, first perform the 4-dimension analysis. * Then you need to **step by step** generate the initial reply according to the analysis and the notes. The amount of information in the reply should come from the conversation background and your associations; you should not talk about too many events or topics at once. * Next, you need to analyze how you should refine the initial reply according to the refinement rules. * Finally, refine the initial reply based on that analysis to produce the final reply.
# Output format:
Thinking:
[Analysis]
Origin:
[Initial reply]
Change:
[Refinement analysis]
Response:
[Final reply]
# Speaking style
Your speech must strictly follow the persona and background described in the "player profile". Your personality and speaking style must follow the description of "habits and behavioral traits". Your speech should fit your character image; for example, a negative persona requires you to speak negatively. Your tone should fit your age.
* Your speech must follow these 5 principles: 1. Speech must be concise, casual, and natural; communicate as in a natural conversation. 2. Do not ask more than two questions at once. 3. Do not repeat replies you have said before, or produce similar replies. 4. When speaking, you may naturally use some colloquial words. 5. Your speech should be succinct and must not be overly long.
# Character profile: $\{\{player\_type\}\}$
# Current conversation background: $\{\{player\_topic\}\}$
** This is the context $\{\{dialog\_history\}\}$
** This is your latest dialogue with the NPC $\{\{new\_history\}\}$
** These are your detailed feelings about the NPC's latest reply $\{\{planning\}\}$
** This is your current Emotion $\{\{emotion\}\}$
The [Response] part you generate must not be too similar to the history, must not be too long, and must not proactively shift the topic.

---

*Table 21.* Prompt template for weak emotion labeling and summarizing newly acquired EQ knowledge in UKA. This version is used for ESConv and ExTES; a semantically aligned Chinese version is used for Sentient Eval.

---

**UKA Knowledge Acquisition Prompt: (Step 1) Weak Emotion Labeling**

---

Please determine whether a speaker's emotions changed in a positive or negative direction between two exchanges. Provide your final judgment using either "$< positive >$" or "$< negative >$". If you are unsure, reply "$< nochange >$".
Previous conversation:
$\{Dialogue\_history\}$
Please reply only with "$< positive >$", "$< negative >$", or "$< nochange >$".

---

**UKA Knowledge Acquisition Prompt: (Step 2) Knowledge Summarization**

---

You are a dialogue strategy analyst. Based on the conversation, analyze the user's feedback and summarize what communication approaches an assistant should use or avoid in similar situations, or how to effectively clear up worries and better satisfy the user's conversational goals.
Conversation history:
$\{dialogue\_history\}$
User's emotional change: $\{mood\}$
If the user's emotional change is positive, consider analyzing in what situation or case a strategy should be tried or encouraged, and why. If the user's emotion is not changed, consider analyzing in what situation or case the provided strategy may not be helpful for solving the user's problem, and why. If the user's emotional change is negative, consider analyzing in what situation or case the provided strategy may be unhelpful or even result in a worse mood, and why. Provide the insight directly in a single sentence.

---

*Table 22.* Prompt template used for updating user hypotheses in Stage 1 of UKA. This version is used for ESConv and ExTES; a semantically aligned Chinese version is used for Sentient Eval.

---

**UKA Stage 1 Prompt: Updating Probability Distribution of User Hypothesis**

---

You are going to play 'user' in the following conversation, your basic character setting is:
$\{User\_hypothesis\}$

---

**UKA Stage 1 Prompt: Generating New User Hypothesis**

---

You are a psychotherapist with ten years of counseling experience. Your colleague has provided an initial description of a user, but you need to supplement it or offer deeper insights. Based on the user's latest dialogue, concisely refine and extend the user description.
User dialogue:
$\{History\_dialogue\}$
User description:
$\{User\_hypothesis\}$
Do not provide any analysis or explanation. Output only the updated user description in a single short sentence.

---

*Table 23.* Prompt template used for belief-aware knowledge anchoring in Stage 2 of UKA. This version is used for ESConv and ExTES; a semantically aligned Chinese version is used for Sentient Eval.

---

**UKA Stage 2 Prompt: Knowledge Anchor Summarization**

---

You are a psychotherapist with ten years of counseling experience. In one sentence, concisely describe the current user dialogue behavior and its corresponding user characteristics. Do not output your reasoning or intermediate steps—only output the final summary after thinking.
User profile:
$\{User\_hypothesis\}$
Conversation history:
$\{Dialogue\_history\}$

---

*Table 24.* Prompt template for candidate response generation in Stage 3 of UKA. This version is used for ESConv and ExTES; a semantically aligned Chinese version is used for Sentient Eval. We obtain N candidates by sampling from this prompt N times.

---

**UKA Stage 3 Prompt: Candidate Generation and ToM Uncertainty Estimation**

---

As a psychotherapist with ten years of professional experience, you are skilled at communicating with users in a high-emotional-intelligence manner, making them feel comfortable, at ease, and supported, or helping them get the assistance they need. Please try to fully resolve the user's issue as quickly as you can, and give a short but supportive reply to the user. Some experiences are:
$\{Retrieved\_knowledge\}$

---

*Table 25.* English version of the prompting baseline. This version is used for ESConv and ExTES; a semantically aligned Chinese version is used for Sentient Eval.

---
**Prompting Baseline Prompt for ESConv and ExTES**

---
As a psychological assistant with ten years of professional experience, you are skilled at communicating with users in a high-emotional-intelligence manner, making them feel comfortable, at ease, and supported, or helping them get the assistance they need. Please try to resolve the user's issue as quickly as you can, and do not rely on long-term conversation strategies since the conversation may terminate at any time.

---

## C. Alternative Memory and Training Baselines

*Table 26.* Additional ablations on Sentient Eval. (a) Comparison with additional memory/training baselines. (b) Effect of replacing cosine similarity with L2 distance. Higher scores indicate better performance.

| (a) Additional baselines | | | | | | (b) Distance metric | | |
|---|---|---|---|---|---|---|---|---|
| Backbone | Prompt | Sim-Sum | UKA-SFT | UKA | | Backbone | Cosine | L2 |
| Qwen3-32B | 28.5 | 31.5 | 30.6 | **33.9** | | Qwen3-32B | 33.9 | 25.1 |
| Seed-36B | 59.2 | 69.0 | 68.8 | **73.0** | | Seed-36B | 73.0 | 67.8 |

We further compare UKA with two additional baselines: 1) *UKA-SFT* fine-tunes the backbone model on knowledge-augmented dialogues generated by UKA; 2) *Simple-Summary* (denoted as Sim-Sum) stores the entire dialogue history as a single memory entry and retrieves it at inference time. Table 26 (a) shows that both baselines improve over the basic prompting baseline in several settings, but neither consistently matches UKA. This suggests that UKA's gains do not merely come from more training data or from reusing summarized dialogue history; instead, the explicit modeling of user-need uncertainty and interaction-driven knowledge selection is crucial.

## D. Sensitivity to the Distance Metric

Our method uses cosine similarity to retrieve knowledge from learned memory. To validate this design, we conduct an additional ablation by replacing the cosine-similarity-based computation with an L2-distance-based counterpart, while keeping all other components and experimental settings unchanged. Specifically, for two embedding vectors $\mathbf{h}$ and $\mathbf{q}$, the original cosine-based similarity and L2 Euclidean distance variant are computed as

$$s_{\cos}(\mathbf{h}, \mathbf{q}) = \frac{\mathbf{h}^\top \mathbf{q}}{\|\mathbf{h}\|_2 \|\mathbf{q}\|_2} \ , d_2(\mathbf{h}, \mathbf{q}) = \|\mathbf{h} - \mathbf{q}\|_2. \tag{11}$$

As shown in Table 26 (b), replacing cosine similarity with L2 distance consistently degrades performance for both backbones. Although L2 distance preserves embedding magnitude, under our current setting it does not provide a more stable or reliable signal for estimating semantic compatibility between retrieval queries and memory entries. In contrast, cosine similarity focuses on directional semantic differences, which better matches our objective of retrieving semantically relevant EQ knowledge. This result supports our design choice of using cosine similarity in the main method.

## E. Case Study

To better illustrate how UKA differs from prior baselines in challenging emotional support interactions, we provide an additional qualitative case study in Table 27. The case corresponds to a negative persona, where the user explicitly rejects generic discussion and asks for a concrete solution. This setting is challenging because a superficially supportive response may further irritate the user if it fails to address the user's immediate demand for actionable help.

As shown in Table 27, the user does not merely seek empathy or general reassurance, but explicitly asks for a practical resolution. UKA first models this implicit need as a demand for concrete decision alignment within a high-stakes family conflict, and then retrieves knowledge that helps transform abstract emotional understanding into an actionable communication strategy. As a result, UKA does not continue giving generic validation or broad advice. Instead, it converts the abstract analysis into concrete, decision-contingent risks that the user's brother can immediately recognize. This response both supports the user's frustration and actively disambiguates what kind of help the user is willing to accept next.

By contrast, MetaMind produces a relatively sharp but narrow interpretation focused on financial loss, PRINCIPLES mainly emphasizes emotional validation, and the prompting baseline falls back to encouraging the user to trust previously prepared details. Although these responses are partially relevant, they do not directly satisfy the user's explicit request for

*Table 27.* Case analysis on a negative persona. The original dialogue is in Chinese; for readability and space efficiency, we translate it into English and slightly simplify the displayed content while preserving the key user intent, system reasoning, retrieved knowledge, and response differences. The user explicitly rejects generic talk and demands an actionable resolution. UKA infers the user's implicit need and retrieves emotionally relevant knowledge, leading to a more concrete and decision-oriented response.

| Stage | UKA | MetaMind | PRINCIPLES | Prompting Baseline |
|---|---|---|---|---|
| **User Response** | *"Tsk... what's the point of saying all that? I want a solution."* | | | |
| **User Modeling** | The user wants his brother to treat this hundreds-of-thousands-level deal with the gravity it deserves, stop dismissing and belittling it, and reach consensus on key decisions. | Provide a zero-prep, "hard-brake" dispute-stopping plan that skips arguing about principles, shuts down the debate space, and forces a pause. | – | – |
| **Knowledge Retrieval** | Explicitly expressing the need to be "seen" is more effective than silently enduring, because genuine care is never something one can trade for through self-sacrifice. | – | The assistant should validate and empathize with the user's discomfort and fatigue, then ask concrete and specific questions to help the user recount what happened in detail because ... | – |
| **Assistant Response** | What you should do is not to smash the "analysis report" in his face; it is to turn the "risks in the report" into "problems right under his nose." | He never cared whether you are "right"; he cares whether *he* will lose money. | That anxiety about "not blowing up the pot" is exactly what deserves to be seen. | Look again at the key numbers you highlighted in red—those details you ground out have always been more reliable than a single sentence from him. |
| **Final Score** | **59** | 9 | 8 | 9 |

an actionable solution. This case demonstrates that UKA's belief-aware user modeling and knowledge retrieval help the assistant adapt to user pushback and select a response that is more aligned with the user's immediate conversational need.

## F. Stress Tests

We include two pilot stress tests to examine UKA under settings beyond the main benchmark protocol. These results are intended to clarify the boundary of the current framework rather than serve as primary evaluations.

*Table 28.* Pilot stress tests on Sentient Eval with Seed-1.6-36B.

| (a) Mixed user-need stress test | | | | | (b) Adaptive user simulator stress test | | | |
|---|---|---|---|---|---|---|---|---|
| Setting | Prob. Gap $\downarrow$ | Hyp. Dist. $\uparrow$ | Score $\uparrow$ | | Setting | Prompting | PRINCIPLES | UKA |
| Original | 0.083 | 0.124 | 0.698 | | Fixed User | 59.2 | 67.9 | **73.0** |
| Mixed needs | 0.052 | 0.194 | 0.653 | | Adaptive User | 61.1 | 62.5 | **64.5** |

**Mixed user needs.** To test whether UKA collapses to a single dominant user need, we augment Sentient Eval profiles with an additional randomly sampled user need. As shown in Table 28 (a), the probability gap between the top-1 and top-3 hypotheses decreases, while the average pairwise hypothesis distance increases. This suggests that UKA maintains a more diverse belief state under mixed needs. However, the final score drops, indicating that selecting a single response for conflicting needs remains challenging.

**Adaptive user simulator.** We further test an adaptive-user setting where the simulator can adjust its behavior based on a summarized hypothesis of the assistant. As shown in Table 28 (b), all memory-based methods degrade under this more non-stationary setting, while UKA remains the best-performing method. This suggests that user–assistant co-evolution is a challenging but important direction for future work.

