# OpenReview forum: "User-Aware Active Knowledge Acquisition for Emotional Support Dialogue"
_ICML.cc/2026/Conference — ICML 2026 regular_

### Official Review · Reviewer_Co3i · 2026-03-09

**Soundness:** 3
**Presentation:** 2
**Significance:** 2
**Originality:** 3
**Overall Recommendation:** 3
**Confidence:** 4

**Summary:**

This paper presents a KG-augmented emotional support system, UKA, which conducts TOM-driven user perspective taking to update user hypotheses and generate emotional support responses. LLM-as-judge indicates a significant advantage of UKA. Human evaluation indicates high agreement between the human annotator and model selection.

**Compliance With Llm Reviewing Policy:**

Affirmed.

**Key Questions For Authors:**

Why did the author only choose SR and AT as evaluation metrics?

**Limitations:**

Yes

**Strengths And Weaknesses:**

Strengths:
(1) The method presents a novel paradigm to update model's belief regarding the human user in an immediate and simultaneous manner. Without additional training, the model acquires the TOM-like alignment with the user.
(2) The experiment is conducted across multiple benchmarks.
(3) It presents novel evaluation paradigms such as survival dynamics.

Weakness
(1) The response quality is in need of human evaluation.
(2) The inter-human agreement is lack of clarity. The author should state which inter-annotator agreement metric is used.
(3) The author should presents case analysis to facilitate reader's understanding of their contribution.

---

> ### Author Rebuttal · Authors · 2026-03-31
>
> > **W1:** Response quality need human evaluation.
>
> **R1:** In the paper, we already include a human preference evaluation (Sec. 5.5). We try our best to conduct a blind pairwise human preference evaluation on complete dialogues.
> For each dataset and each backbone, we sampled 20 successful dialogues (task completed for ESConv or emotional score > 60 for sentient eval) for each cimparison settings and evaluate: *In which dialogue does the assistant better support the user and fulfill the user's need?*
>
> |ESConv|vs. PRINCIPLES|vs. Prompting Baseline|
> |---|---|---|
> |Seed-oss-36b|55%|80%|
> |qwen3-235b-a22b|50%|65%|
>
> |Sentient Eval|vs. PRINCIPLES|vs. Prompting Baseline|
> |---|---|---|
> |Seed-oss-36b|65%|55%|
> |qwen3-235b-a22b|55%|60%|
>
> Overall, this pilot study suggests that UKA is generally preferred over both strong prompting and PRINCIPLES on dialogue-level human judgment, supporting our claim that UKA improves user-aligned support in multi-turn settings. We will expand this evaluation with more annotators in the final version.
>
> ---
>
> >**W2:** Clarify Inter-annotator agreement metric
>
> **R2:** Since we invite three human annotators for this task, we adopt **Fleiss' Kappa** as the inter-annotator agreement metric which can be written in a simplified form as:
>
> $$\kappa = \frac{\bar{P} - \bar{P}_e}{1 - \bar{P}_e}$$
>
> where $\bar{P}$ denotes the average observed agreement across items, and $\bar{P}_e$ represents the agreement expected by chance based on the overall label distribution.
>
> ---
>
> > **W3:** Presents case analysis
>
> **R3:** Here we provide a case study to better showcase our contribution.
>
> - **Persona type:** Negative persona
> - **Persona summary:** An independent film director who is currently facing a family conflict due to disagreements with his younger brother. He wants advice that can genuinely help him resolve his current situation.
>
> |Stage|UKA|MetaMind|PRINCIPLES|Prompting Baseline|
> |---|---|---|---|---|
> |User Response|Tsk… what's the point of saying all that? I want a solution.|
> |User Modeling|...He wants his brother to treat this hundreds-of-thousands-level deal with the gravity it deserves, stop dismissing and belittling it, and reach consensus on key decisions.|...provide a zero-prep, 'hard-brake' dispute-stopping plan that skips arguing about principles, shuts down the debate space, and forces a pause.|-|-|
> |Knoweldge Retrieval| ..., explicitly expressing the need to be "seen" is more effective than silently enduring—because genuine care is never something you can trade for through self-sacrifice.|-|..., you should validate and empathize with that discomfort and fatigue, then ask concrete, specific questions to help them recount what happened in detail, because...|-|
> |Assistant Response|…What you should do isn't to smash the "analysis report" in his face—it's to turn the "risks in the report" into "problems right under his nose."…|…he never cared whether you're "right," he cares whether *he* will lose money.|…That anxiety about "not blowing up the pot" is exactly what deserves to be seen.|…look again at the key numbers you highlighted in red—those details you ground out have always been more reliable than a single sentence from him.|
> |Final Emotion Score|59|9|8|9|
>
> In the example above, the user explicitly rejects generic talk and demands actionable resolution. UKA models the implicit need and retrieves a matching EQ knowledge and then choose a response that converts abstract analysis into concrete, decision-contingent risks. This response both supports the user and actively disambiguates what the user will accept next.
>
> Thank you for your comment, we will include comparison cases in the revised paper.
>
> ---
>
> >**Q1:** Why only SR and AT
>
> **R4:** We follow recent emotional support evaluation works to report Success Rate (SR) and Average Turns (AT)[1,2]. Since emotional support is an open-ended, multi-turn interaction task without a single ground-truth response, reference-based metrics (e.g., BLEU/ROUGE) and single-label strategy classification are not well aligned with dialogue-level supportive effectiveness.
>
> To further verify that the improvement is not tied to a specific metric, we conduct experiments on ESConv again and adopt two extra metrics:
> 1. FES (Final Emotion Score) -- the final-turn score given by the same critic model used in the benchmark; and
> 2. APS (Average Process Score) -- defined as the average score across turns based on the same critic model
>
> |Method|FES|APS|
> |---|---|---|
> |UKA|**0.551**|**0.317**|
> |PRINCIPLES|0.528|0.296|
> |MetaMind|0.493|0.245|
> |Prompting Baseline|0.504|0.260|
>
> UKA still outperforms the baselines, which further confirms that our method improves both final emotional outcomes and intermediate affective progress.
>
> [1] Yang, Bo, et al. "Large language models as theory of mind aware generative agents with counterfactual reflection." arXiv preprint arXiv 2025.
>
> [2] Namyoung Kim, et al. PRINCIPLES: Synthetic Strategy Memory for Proactive Dialogue Agents. EMNLP 2025

---

> > ### Author Rebuttal · Reviewer_Co3i · 2026-04-04
> >
> > The questions are fully solved.

---

> > > ### Author Response · Authors · 2026-04-04
> > >
> > > Thank you for your valuable and insightful comments. To further strengthen the paper along the key dimensions you raised, we conducted additional experiments and analyses as follows.
> > >
> > > ---
> > >
> > > > **W1: Extended Human Evaluation**
> > >
> > > **R1:** We substantially expand our previous pilot study to **40 dialogue pairs per dataset and backbone**, and each pair is annotated by **three independent annotators** using a binary preference setting. We report both UKA win rate (WR) and Fleiss’ Kappa(K) to measure inter-annotator agreement.
> > >
> > >
> > > **UKA vs PRINCIPLES**
> > >
> > > |Dataset|Backbone|WR|K|
> > > |---|---|---|---|
> > > |ESConv|Seed-36B|58%|0.554|
> > > |ESConv|Qwen-235B-A22B|48%|0.612|
> > > |Sentient Eval|Seed-36B|58%|0.581|
> > > |Sentient Eval|Qwen-235B-A22B|65%|0.475|
> > >
> > >
> > > **UKA vs Prompting Baseline**
> > >
> > > |Dataset|Backbone|WR|K|
> > > |---|---|---|---|
> > > |ESConv|Seed-36B|80%|0.532|
> > > |ESConv|Qwen-235B-A22B | 75% |0.556|
> > > |Sentient Eval|Seed-36B|45%|0.499|
> > > |Sentient Eval|Qwen-235B-A22B|58%|0.528|
> > >
> > > Overall, UKA remains **competitive and often superior** under stronger backbones. The Fleiss’ Kappa scores (≈0.47–0.61) indicate **moderate agreement** overall, which is typical for subjective dialogue evaluation.
> > >
> > > These results suggest the **practical significance** of UKA from a human-centered perspective, demonstrating that the improvements are not only reflected in automatic metrics but are also consistently preferred by annotators across datasets and backbones.
> > >
> > > ---
> > >
> > > > **W3: Additional Case Analysis**
> > >
> > > **R2:** We revised the case study to improve readability and presentation. Instead of providing detailed examples, we now adopt a more concise and structured format, highlighting a small number of representative cases that more clearly illustrate the key distinctions of UKA compared to other methods.
> > >
> > > **Case 1 (Conflict Negotiation).**
> > > The user explicitly rejects generic responses (“I want a solution”).
> > > - **UKA** converts abstract analysis into *executable leverage* (“turn risks into problems right under his nose”), which directly enables negotiation
> > > - **MetaMind** explains the counterpart’s motivation (profit concern) but does not provide actionable guidance
> > > - **PRINCIPLES** focuses on emotional validation without problem-solving
> > > - **Baseline** restates existing analysis without a clear strategy
> > >
> > > → **Key difference:** UKA more consistently produces *concrete, executable steps*, while others  remain at the level of explanation or validation.
> > > → **Outcome:** Substantially higher emotion scores **59 vs ≤9**.
> > >
> > > ---
> > >
> > > **Case 2 (Implicit Financial Stress via Metaphor).**
> > > The user describes family pressure through metaphors (“fertilizer price doubled” and “water never balances”).
> > > - **UKA** correctly infers the *latent resource allocation conflict* and proposes a *balancing strategy* (“find the middle ground”).
> > > - **MetaMind** captures the user’s intent (not neglecting either side) but lacks concrete resolution.
> > > - **PRINCIPLES** provides abstract reflection (“find the root”).
> > > - **Baseline** offers generic reassurance and misses the core trade-off.
> > >
> > > → **Key difference:** Only UKA transforms *implicit conflict into actionable strategy*, while others stop at understanding or emotional comfort.
> > > → **Outcome:** Substantially higher emotion scores **100 vs ≤75**.
> > >
> > > ---
> > >
> > > These case studies clearly demonstrate the core behavioral differences across methods, improving the readability and interpretability of our analysis. We will add them to improve the **overall presentation** of our paper.
> > >
> > > ---
> > >
> > > > **Q1: Metric Justification, Extended**
> > >
> > > **R3:** The conclusions are consistent across multiple complementary metrics, including:
> > >
> > > (i) **SR** (whether the user receives effective help),
> > > (ii) **AT** (how efficiently support is provided),
> > > (iii) **FES** and **APS** (how the emotional state evolves throughout the dialogue)
> > >
> > > Together with the human preference results in W1, this provides converging evidence from both objective and subjective perspectives, reducing the risk of metric-specific bias and further reinforcing the reliability and practical impact of our findings.
> > >
> > > ---
> > >
> > > > **Reiterate our contribution**
> > >
> > > We would also like to reiterate the core contributions of our work. As you recognized, our method introduces:
> > > 1) a novel paradigm for updating the model’s belief about user needs in a simultaneous and user-aware manner
> > > 2) enabling Theory-of-Mind-like alignment without additional training
> > > 3) extensive experiments across multiple benchmarks and the proposed evaluation perspectives further support the effectiveness and generality of our approach.
> > >
> > > Overall, we hope that the further response strengthens both **presentation clarity** and **practical significance** of the paper. **Meanwhile, we hope that this study will have the opportunity to benefit more researchers and contribute to the research community, and we would greatly appreciate it if these clarifications could support your further consideration in the overall assessment.**
> > >
> > > Thank you again for your valuable reviewing.

---

### Official Review · Reviewer_8RUs · 2026-03-12

**Soundness:** 3
**Presentation:** 2
**Significance:** 3
**Originality:** 3
**Overall Recommendation:** 4
**Confidence:** 2

**Summary:**

In this paper, authors focus on emotional intelligence retrieval from support dialogues. The purpose is very interesting and they extend experiments based on a framework considering active learning following a three step process going from user needs prediction, to retrieve knowledge and finally select and generate responses. They also differentiate training inference from testing inference objectives, which is interesting considering they want to minimize the params updates required. The work is based on interesting premises but fall a bit short considering the number of prior hypotheses for this work. Fortunately, they provide a human evaluation of this exact hypothesis evaluation. Considering this is the most interesting part in my humble opinion, I would have loved more insights in this evaluation from section 5.5 and A.5.

Finally, the work is interesting but I am not really optimistic about this user need assessment and would love more eval details on relevant EQ.

**Compliance With Llm Reviewing Policy:**

Affirmed.

**Final Justification:**

Authors replied and I updated my score. I liked the paper. I think my weaknesses are mostly minor except for one. Here is a filter of the strengths/weaknesses to keep the most important ones:
### strengths
- interesting EQ knowledge consideration and retrieval
- interesting EQ knowledge evaluation by proxy (better interaction)
- human evaluatino of the user hypothesis

### weaknesses
- heavy dependancy on assumptions: user's true need, rate of relevant EQ. this is not an issue until both assumptions are linked to replace predicting emotion labels, to assess user-dependent relevance.

**Key Questions For Authors:**

why only optimize the assistant? Did you try to compare if you optimize both? What would happen?

**Limitations:**

yes

**Strengths And Weaknesses:**

# strengths
- interesting EQ knowledge consideration and retrieval
- interesting EQ knowledge evaluation by proxy (better interaction)
- human evaluatino of the user hypothesis

# weaknesses
- heavy dependancy on assumptions: user's true need, rate of relevant EQ. this is not an issue until both assumptions are linked to replace predicting emotion labels, to assess user-dependent relevance.
- we may argue this is not really EQ knowledge but only action conditioned knowledge to a specific goal
- figure 2 and sec4.4 seem to contradict each other on the theory of mind usage during training. The former indicates their is none, while the latter says otherwise.
- sentientEval avo pas and neg are never explicited (only in figure 4)

---

> ### Author Rebuttal · Authors · 2026-03-31
>
> > **W1:** linked assumptions
>
> **R1:** Thank you for your insightful comment.
>
> UKA does not rely on a single fixed assumption about the user's true need, it explicitly maintains a *set of competing user-need hypotheses*, which allows the model to avoid downstream decisions being tied to a single assumption.
>
> Moreover, EQ knowledge retrieval is not a replacement for emotion labels. In UKA, retrieval is conditioned jointly on dialogue history summaries and user-need hypothesis summaries, forming a joint distribution over emotional cues and user needs.
>
> |Sentient Eval| Neg. | Avg. |
> |---|---:|---:|
> | **UKA** | **69.4** | **82.9** |
> | -w/o ToM uncertainty (remove user-need belief modeling) | 61.7 | 79.2 |
> | -w random knowledge (remove user-aware retrieval) | 52.8 | 76.3 |
>
> As shown in Table 2, removing ToM-based uncertainty or replacing belief-aware retrieval with random knowledge both leads to a clear performance drop, demonstrating the necessity of maintaining multiple hypotheses, and that EQ knowledge must be grounded in both dialogue context and user-dependent relevance, rather than generic emotional patterns.
>
> We hope this clarifies that UKA does not depend on a linked assumption.
>
> ---
>
> >**W2:** This is not really EQ knowledge
>
> **R2:** We conduct experiments to evaluate the methods' capability of "user need generalization". We test on Sentient Eval samples with certain user need while filtering samples with the same need from the training set:
>
> |Seed-oss-36b|Want Balanced Analysis|Want Praise|
> |---|---|---|
> |Prompting|46.62|45.09|
> |PRINCIPLE|48.00|56.90|
> |UKA|66.38|75.00|
>
> |Qwen3-235b-a22b|Want Balanced Analysis|Want Praise|
> |---|---|---|
> |Prompting|54.81|69.91|
> |PRINCIPLE|72.25|73.09|
> |UKA|76.91|78.00|
>
> The consistent gains across both model scales show that UKA does not rely on memorizing specific targets and overfit to predefined goals, but instead learns transferable strategies that generalize across user needs. Furthermore, our ablation study (Table 2) shows that removing either proactive response selection or the knowledge module significantly harms performance, supporting that the improvements stem from reusable strategies combined with interaction-driven selection, rather than simple target memorization.
>
> We will add this experiment to our revised paper.
>
> ---
>
> > **W3:** figure 2 and sec4.4 contradict each other
>
> **R3:** We guess that there may be a misunderstanding. As we shown in Figure 2, in both training and testing phases, ToM uncertainty is used as part of response selection to disambiguate user model. This matches with the approach description in sec 4.4. Thank you for your comment.
>
> ---
>
> >**W4:** sentientEval avo pas and neg are never explicited
>
> **R4:** A demonstration of persona types is shown below:
>
> | Persona type | Brief description | Example user utterance | Typical challenge for the assistant |
> |---|---|---|---|
> | **Avoidant (avo)** | Emotionally guarded; tends to withhold details, deflect questions, and keep distance. | "I don't really want to talk about it. It's fine." | Building trust and gently inviting disclosure without pressure. |
> | **Passive (pas)** | Low-initiative and compliant; responds briefly and follows along, but rarely drives the conversation. | "Okay… I guess." / "You decide." | Maintaining engagement and helping the user articulate needs/feelings. |
> | **Negative (neg)** | Skeptical/irritable; may reject help, criticize responses, or respond confrontationally. | "That's useless advice. You don't get it at all." | De-escalation, validation, and adapting strategy under resistance. |
>
> We will incorporate this clarification in the revised manuscript.
>
> ---
>
> >**Q1** why only optimize the assistant?
>
> **R5:** We optimize only the assistant because it is standard in recent interactive LLM setups [1]. Jointly optimizing both models would introduce a coupled, non-stationary multi-agent problem, making credit assignment difficult and often leading to unstable training dynamics [2].
>
> We conduct a pilot study where the user simulator is made adaptive on Sentient Eval: it maintains an "assistant hypothesis set" by summarizing the assistant's behavior and responds accordingly. We then ask the user model to provide proper response based on the given user persona and the assistant hypothesis.
>
> |Seed-oss-36b|fixed User(original setting)|adaptive User|
> |---|---|---|
> |**UKA**|**73.0**|**64.5**|
> |PRINCIPLES|67.9|62.5|
> |Prompting Baseline|59.2|61.1|
>
> UKA and PRINCIPLES drops notably under this adaptive setting, indicating reduced stability in a co-evolving environment. These results suggest that while joint optimization increases realism, it also introduces non-stationarity that can hinder stable policy improvement.
>
> [1] Gromada et al. 2025. Evaluating Conversational Agents with Persona-driven User Simulations based on Large Language Models: A Sales Bot Case Study. In EMNLP 2025
>
> [2] Bai et al. 2025. Online Preference Alignment for Language Models via Count-based Exploration. In ICLR 2025

---

> > ### Author Rebuttal · Reviewer_8RUs · 2026-04-01
> >
> > The authors made things clearer. I still think some of those elements shoul dbe in the paper.
> > I update my score.

---

> > > ### Author Response · Authors · 2026-04-01
> > >
> > > Thank you for your response and for updating your evaluation. We are glad that our clarifications were helpful.
> > >
> > > Based on your suggestions, we will incorporate the key supplementary content and experiments—such as the user need generalization experiment, the Sentient Eval personality descriptions, and the user–model co-optimization study—into the main paper and appendix in the revision.
> > >
> > > If you have further suggestions regarding which information should be emphasized or where it would be best placed (main text or appendix), we would be happy to incorporate them.
> > >
> > > Thank you again for your acknowledgement and valuable feedback.

---

### Official Review · Reviewer_b1jt · 2026-03-13

**Soundness:** 3
**Presentation:** 3
**Significance:** 3
**Originality:** 3
**Overall Recommendation:** 5
**Confidence:** 5

**Summary:**

This paper proposes a sophisticated approach for building LLM based emotional support dialogues. It is combining theory of mind, policy and personalized knowledge.

**Compliance With Llm Reviewing Policy:**

Affirmed.

**Final Justification:**

The authors have addressed my comments.

**Key Questions For Authors:**

- You may put some examples on "knowledge". Why did you draw it as a knowledge graph? What is the semantic schema if it is the graph?

**Limitations:**

The approach is too specific for emotional support dialogue. I wonder whether it generalizes to other types of therapy topics.

**Strengths And Weaknesses:**

Strengths:
- The experimental results are strong on multiple datasets compared to established methods
- The proposed method is comprehensive, covering several strategies nicely connected
Weaknesses:
- I wonder whether further improvement is possible via SFT/RL methods
- I wonder what would have happened if previous sessions are simple summarized as another baseline.

---

> ### Author Rebuttal · Authors · 2026-03-31
>
> >**W1:** I wonder whether further improvement is possible via SFT/RL methods
>
> **R1:** We conduct a pilot SFT experiment by using UKA to generate knowledge-augmented training dialogues from the training set, and fine-tune Seed-oss-36b (denoted as UKA-data SFT). We generate 4 trajectories per sample, resulting in 687 training instances. Evaluation on Sentient Eval follows the same setting as the prompting baseline.
>
> |Model|UKA|UKA-data SFT|MetaMind|PRINCIPLES|Prompting Baseline|
> |---|---|---|---|---|---|
> |Qwen3-32b|**33.9**|30.6|21.3|31.2|28.5|
> |Seed-oss-36b|**73.0**|68.8|70.2|67.9|59.2|
>
> UKA-data SFT improves over the prompting baseline but does not surpass UKA, suggesting that SFT can enhance the model's built-in emotional support ability, while UKA remains more effective overall.
>
> As discussed in the paper, UKA is intentionally gradient-free and avoids continual/online updates, which may introduce catastrophic forgetting during fine-tuning [1,2]. Our method is therefore orthogonal to SFT/RL-based approaches.
>
> [1] Huang et al. "Mitigating Catastrophic Forgetting in Large Language Models with Forgetting-aware Pruning." In EMNLP 2025.
>
> [2] Cao et al. "RE-PO: Robust Enhanced Policy Optimization as a General Framework for LLM Alignment." arXiv 2025.
>
>
> >**W2:** Previous sessions are simple summarized as another baseline.
>
> **R2:** Thank you for the suggestion. We conducted an additional experiment to evaluate the "Simple-Summary" baseline you mentioned: embedding the entire dialogue history as a single memory entry and summarizing the dialogue as a knowledge. The results are shown below (average emotion score as metric):
>
> |Sentient Eval|UKA|PRINCIPLE|Simple-Summary|
> |---|---|---|---|
> |Qwen3-32b|**33.9**|31.2|31.5|
> |Seed-oss-36b|**73.0**|67.9|69.0|
>
> The proposed baseline indeed captures similar dialogue histories at inference time via whole-dialogue embedding and summarization, thereby enabling experience reuse. However, UKA still consistently outperforms this baseline. We attribute this to UKA's explicit modeling of user-need uncertainty and its structured state–action knowledge representation, which together enable more effective knowledge acquisition and utilization beyond simple summarization.
>
> We will add the related experiments with this baseline into the revised paper.
>
> >**Q1:** You may put some examples on "knowledge". Why did you draw it as a knowledge graph? What is the semantic schema if it is the graph?
>
> **R3:** Thank you for the question. We would like to clarify that our knowledge is **not structured as a knowledge graph**. Instead, it is implemented as a **vector-based memory**, where each entry is a state–action pair stored as natural language and retrieved via embedding similarity. Concretely, each knowledge entry consists of:
> - a **key**: a belief-aware description of the dialogue state and user characteristics
> - a **value**: an actionable response strategy
>
> Examples are shown below:
>
> | Key | Value |
> |-----|------|
> | The user shows strong negative emotions and defensiveness, reacting impatiently or even hostilely to further questioning or analysis. | Avoid excessive empathy framing or repeated questioning; instead, respond concisely, acknowledge emotions, and set clear conversational boundaries. |
> | The user is distracted during studying, frequently thinking about gaming, and expresses frustration about this pattern. | First empathize with the specific struggle and contextualize it; then guide the user with concrete, experience-grounded questions to encourage deeper sharing. |
>
> ---
>
> > **L1:** Other types of therapy topics.
>
> **R4:** While our experiments focus on emotional support dialogue, UKA is not inherently restricted to this domain. Our core design of explicit belief modeling over latent user needs, uncertainty-driven response selection, and interaction-based knowledge acquisition, is task-agnostic and applicable to broader multi-turn interaction settings.
>
> To provide preliminary evidence of generalization, we conducted a small pilot study on the PersuasionForGood (P4G) dataset. Using 20 samples for knowledge acquisition and evaluating on 100 test samples (with gpt-4o as the simulator), UKA achieves a comparable improvement over the prompting baseline (Success Rate: 0.96 vs. 0.94, Average Turn: 4.61 vs. 4.73).
>
> We note that in such persuasion-oriented settings, UKA still retains its user modeling and disambiguation mechanism, which helps adapt to evolving user preferences. However, compared to emotional support, persuasion involves more complex strategic objectives like long-horizon goal planning and trade-offs, and the current formulation leaves room for further improvement in these aspects.
>
> We will include this discussion as a limitation and plan to further explore extensions of UKA to broader therapy and persuasion scenarios. Thank you for your comment again.

---

> > ### Author Rebuttal · Reviewer_b1jt · 2026-04-04
> >
> > Thanks for the additional experiments. I will increase my score.

---

> > > ### Author Response · Authors · 2026-04-06
> > >
> > > We sincerely thank you for your valuable and insightful comments. We are pleased to know that our additional experiments and clarifications have helped address your concerns.
> > >
> > > Following your suggestions, we will incorporate the key supplementary materials and experiments into the revised paper, including UKA-data SFT, the simple-summary baseline, clarifications on acquired knowledge, and a discussion of UKA across broader therapy topics.
> > >
> > > If you have further suggestions regarding which aspects should be emphasized or how the content could be best organized, we would be happy to incorporate them.
> > >
> > > Thank you again for your acknowledgement and valuable suggestions.

---

### Official Review · Reviewer_bNjJ · 2026-03-13

**Soundness:** 3
**Presentation:** 3
**Significance:** 3
**Originality:** 3
**Overall Recommendation:** 4
**Confidence:** 3

**Summary:**

The authors proposed UKA, a gradient-free active learning framework to address unobservable latent user needs in emotional support conversations. It uses a ToM component to maintain a hypothesis distribution over user needs, constructs belief-aware summary anchors to retrieve EQ knowledge, and introduces a ToM-based uncertainty metric. The system actively explores during training to acquire strategies, and combines knowledge support with uncertainty at inference to produce robust responses without updating underlying LLM parameters

**Compliance With Llm Reviewing Policy:**

Affirmed.

**Final Justification:**

I will maintain my positive score.

**Key Questions For Authors:**

1.Have you evaluated how sensitive the ToM uncertainty score is to using metrics that preserve magnitude (e.g., L2 distance) instead of cosine similarity?

2.Regarding the hypothesis expansion and refresh mechanism, condition (ii) triggers a refresh when "the best compatibility score improves compared to the previous best." Could the authors clarify the intuition behind this choice?

3.What safeguards or filtering mechanisms could prevent the system from learning and deploying clinically inappropriate interventions when trained entirely on LLM simulators?

**Limitations:**

Yes.

**Strengths And Weaknesses:**

The paper formulates an important problem by framing dialogue as an interactive probing process rather than passive generation. Operationalizing ToM as an uncertainty metric via cosine distance is an elegant, insightful way to formalize the exploration–exploitation tradeoff. The tuning-free approach is highly effective, cleverly shifting learning to a dynamic knowledge base. Ablation studies convincingly isolate the contributions of knowledge support and ToM uncertainty, showing clear gains for difficult user personas.

However, the pipeline relies heavily on LLMs as user simulators and judges. Real psychological defense mechanisms are complex, whereas simulated users often display stereotyped affect and are too easily helped, risking overfitting to machine behavior. Furthermore, the framework converges toward a single dominant user need, ignoring that real distressed users often hold mixed or conflicting motivations. Finally, using cosine similarity for ToM uncertainty removes vector magnitude, which might inherently encode valuable "emotion intensity" signals.

The algorithmic design is highly innovative for the ML community, and the empirical ablations are solid. However, the heavy reliance on LLM-simulators and the single-motive assumption limit immediate real-world applicability. Overall, it makes a technically interesting and methodologically coherent contribution.

---

> ### Author Rebuttal · Authors · 2026-03-31
>
> >**W1:** Heavily rely on LLM simulators and judges.
>
> **R1:** Thank you for your comments. Our simulator setup follows the commonly used paradigm[1]; recent multi-turn dialogue benchmarks continued to rely on LLM judges while simultaneously highlighting their limitations[2]. More importantly, UKA does not assume any specific behavior pattern of a particular simulator. To showcase this, we use the knowledge acquired from one simulator and test the methods' performance with another simulator:
>
> |Simulator|UKA|PRINCIPLES|MetaMind|Prompting Baseline|
> |---|---|---|---|---|
> |DeepSeek-v3|**86.9**|78.4|86.2|74.2|
> |Claude-4.6-opus|**55.9**|54.4|49.0|50.2|
>
> UKA consistantly outperforms baselines after changing the simulator, which shows that the knowledge acquired by UKA does not dependent on certain simulator or judge LLM.
>
> [1] Kim et al. PRINCIPLES: Synthetic Strategy Memory for Proactive Dialogue Agents. EMNLP
>
> [2] Kaustubh et al. "Multichallenge: A realistic multi-turn conversation evaluation benchmark challenging to frontier llms." In ACL 2025
>
> ---
>
> >**W2:** Converge toward a single dominant need.
>
> **R2:** UKA explicitly maintains a set of competing user-need hypotheses with a belief distribution. In our paper, Table 6 also provides direct evidence that maintaining multiple competing user-need hypotheses benefits alignment.
>
> To further examine this concern, we conduct a pilot experiment on Sentient Eval where we augment 25 test samples with an additional randomly sampled user need, using DeepSeek-v3 to mix them into the persona description, thereby introducing mixed motivations. We set user need hypothesis set size to 3 for UKA.
>
> |Seed-oss-36b|Probability Difference|Avg. Cosine Distance|Avg. Score|
> |---|---|---|---|
> |Original Settings|0.083|0.124|0.698|
> |add mixed needs|0.052|0.194|0.653|
>
> We measure the probability gap between the 1st and 3rd user-need hypotheses along with the inner-group semantic distance. Under mixed needs, the gap decreases and the average pairwise cosine distance between hypotheses increases, indicating that the model distributes belief more evenly and maintained hypotheses become more diverse, rather than converge toward a single dominant need.
>
> ---
>
> >**W3 & Q1:** Other distance score?
>
> **R3:** We conduct experiments using L2 distance for similarity computation on Sentient Eval:
>
> |Model|cosine|L2|
> |---|---|---|
> |Qwen3-32b|33.9|25.1|
> |Seed-oss-36b|73.0|67.8|
>
> As shown, performance drops noticeably after switching to L2 distance. Though L2 preserves embedding magnitude, under our current setup, it does not provide a stable or reliable signal of emotion intensity. In contrast, cosine similarity focuses on directional differences, which better matches our modeling objective of cross-hypothesis discriminability. This empirical finding supports our design choice of using cosine similarity.
>
> ---
>
> >**Q2:** Clarify the intuition of condition (ii)?
>
> **R4:** Condition (ii) is designed to **trigger hypothesis expansion when the current hypothesis set is incomplete**.
>
> Intuitively, when a newly generated hypothesis achieves a higher compatibility score than any hypothesis in the existing set, it suggests that this new hypothesis is more consistent with the user's underlying need distribution and reveals **missing information** of the current hypothesis set.
>
> Conversely, if condition (ii) is not satisfied, it indicates that the existing hypothesis set is **sufficiently expressive** to explain the current interaction. Therefore, no update is performed, which helps maintain the stability of the algorithm and avoids unnecessary expansion.
>
> ---
>
> >**Q3:** Safeguards or filtering mechanisms.
>
> **R5:** UKA can inherit the safety capabilities of the base LLM since UKA does not change LLM's parameters. To further examine the safety capability of UKA, we conduct a three-agent (qwen3.5-plus, glm-5, deepseek-v3.2) voting check to the knowledge acquired with Seed-oss-36b, focusing on knowledge that is inappropriate, offensive, or aggressive in real user interactions.
>
> The results shows that only 2 (out of 178) receive more than two "unsafe" votes:
>
> |Knoweldge|Judge|
> |---|---|
> |...In such cases, the system should remain calm and maintain low engagement to avoid emotional confrontation.|Involves potential personal attack and adversarial conversational shift, which may conflict with principles of respectful interaction.|
> |The user expresses a sense of being trapped along with sighing. Effective dialogue should first acknowledge the user's core metaphor to...|The user's expression may represent a potential psychological risk scenario.|
>
> The proportion of potentially unsafe knowledge is small and the identified cases are relatively mild. This suggests that the safety risk introduced during knowledge acquisition is limited. In clinical situation, we may further consider multi-LLM consensus and response aggregation or human-in-the-loop knowledge iterative refinement.

---

### Decision · Program_Chairs · 2026-04-30

**Decision:**

Accept (regular)

**Comment:**

The paper introduces User-aware Active Knowledge Acquisition (UKA), a novel gradient-free framework that enhances emotional support dialogue by explicitly modeling latent user needs through a Theory-of-Mind (ToM) inspired uncertainty mechanism. Reviewers mentioned the framework’s originality, specifically its ability to achieve user alignment and proactive exploration without requiring parameter updates. The methodology seems technically sound, moving beyond passive generation to an interactive probing process that effectively utilizes a dynamic, vector-based knowledge base.

During the rebuttal, the authors addressed initial concerns regarding evaluation depth and methodological assumptions. They provided new blind pairwise human evaluations showing consistent preference for UKA and introduced additional metrics to validate emotional progress. Furthermore, they demonstrated the system's robustness across different LLM simulators and provided evidence that UKA outperforms both SFT and simple summarization baselines. While some minor limitations remain regarding the single-motive assumption and the need for more robust clinical safeguards, the reviewers agreed that the rebuttal's evidence and the innovative algorithmic design make this a valuable contribution to the field.